# The m6A reader PRRC2A is essential for meiosis I completion during spermatogenesis

Xinshui Tan[1,2,5], Caihong Zheng[3,5], Yinghua Zhuang[2], Pengpeng Jin[2] & Fengchao Wang [1,2,4] ✉

N6-methyladenosine (m6A) and its reader proteins YTHDC1, YTHDC2, and YTHDF2 have been shown to exert essential functions during spermatogenesis. However, much remains unknown about m6A regulation mechanisms and the functions of specific readers during the meiotic cell cycle. Here, we show that the m6A reader Proline rich coiled-coil 2A (PRRC2A) is essential for male fertility. Germ cell-specific knockout of *Prrc2a* causes XY asynapsis and impaired meiotic sex chromosome inactivation in late-prophase spermatocytes. Moreover, PRRC2A-null spermatocytes exhibit delayed metaphase entry, chromosome misalignment, and spindle disorganization at metaphase I and are finally arrested at this stage. Sequencing data reveal that PRRC2A decreases the RNA abundance or improves the translation efficiency of targeting transcripts. Specifically, PRRC2A recognizes spermatogonia-specific transcripts and downregulates their RNA abundance to maintain the spermatocyte expression pattern during the meiosis prophase. For genes involved in meiotic cell division, PRRC2A improves the translation efficiency of their transcripts. Further, co-immunoprecipitation data show that PRRC2A interacts with several proteins regulating mRNA metabolism or translation (YBX1, YBX2, PABPC1, FXR1, and EIF4G3). Our study reveals post-transcriptional functions of PRRC2A and demonstrates its critical role in the completion of meiosis I in spermatogenesis.

N6-methyladenosine (m6A), the most prevalent modification of eukaryotic mRNA, post-transcriptionally regulates many physiological and pathological processes[1]. The dynamics of m6A is coordinated by the interplay between the m6A "writer" and "eraser" proteins. In mammalian cells, the methyltransferase complex comprising METTL3 and METTL14 acts as the writer to catalyze the m⁶A methylation onto RNAs[2,3]. Conversely, erasers FTO[4] and ALKBH5[5] remove these m⁶A modifications. Many reader proteins have been discovered recently, including YTH family proteins[6–13], HNRNP family proteins[14–16], IGF2BPs[17], eIF3[18], FMRP[19], SND1[20], and PRRC2A[21]. Reader proteins generally act as functional mediators to regulate the splicing, transportation, stability/degradation, storage, and translation status of m6A-modified RNAs[22]. In a previous study, PRRC2A was demonstrated as an m6A reader and regulates oligodendroglial specification in brain development by promoting the stability of *Olig2* mRNA[21].

During spermatogenesis, male germ cells develop from spermatogonia to spermatocytes, and then to spermatids[23]. In prophase I, germ cells undergo pairing and recombination of homologous

[1]Graduate School of Peking Union Medical College, Chinese Academy of Medical Sciences, Beijing, China. [2]National Institute of Biological Sciences, Beijing, China. [3]Key Laboratory of Genomic and Precision Medicine, Beijing Institute of Genomics, Chinese Academy of Sciences, and China National Center for Bioinformation, Beijing 100101, China. [4]Tsinghua Institute of Multidisciplinary Biomedical Research, Tsinghua University, Beijing 102206, China. [5]These authors contributed equally: Xinshui Tan, Caihong Zheng. ✉e-mail: wangfengchao@nibs.ac.cn

chromosomes. Based on the appearance of chromosomes, prophase I is divided into four substages named leptotene, zygotene, pachytene, and diplotene[24]. Notably, in pachytene and diplotene spermatocytes, transcription on X and Y chromosomes is silenced, which is called meiotic sex chromosome inactivation (MSCI)[24]. Subsequently, homologous chromosomes and sister chromosomes separate during meiosis I and meiosis II, respectively[23]. Similar to somatic cells, the centrosome functions in regulating spindle and chromosome behaviors during the metaphase of male meiosis[25,26]. Proper chromosome alignment is dependent on normal kinetochore-microtubule attachment and spindle organization[27]. When the kinetochore-microtubule attachment or chromosome alignment is disrupted, the spindle assembly checkpoint (SAC) was activated and this prevents the metaphase-to-anaphase transition to avoid the formation of aneuploidy gametes during metaphase of meiosis[28].

A previous study showed that m6A-modified transcripts exist in all types of male germ cells[11,29]. Correspondingly, m6A writers, METTL3 and METTL14, were shown to regulate the maintenance of spermatogonial stem cells, spermatogonia differentiation, meiosis initiation, and spermiogenesis[29,30]. And the ablation of the m6A eraser, ALKBH5, leads to the depletion of pachytene spermatocytes and spermatids in testes[5]. FTO, another eraser, modulates the cell cycle in the mouse GC-1 spermatogonial cell line[31]. m6A is required for the proper development at all stages of spermatogenesis. As the functional mediators of m6A, several readers have been reported to regulate spermatogenesis. Specifically, YTHDC2 was shown to regulate meiosis prophase by modulating mRNA abundance and translation efficiency[7,8,11,32,33], but recent studies suggested that the functions of YTHDC2 in spermatogenesis are independent of m6A recognition[34,35]. YTHDC1 is required for the survival and development of spermatogonia[36]. YTHDF2 facilitates cell proliferation and cell adhesion of GC-1 cells by modulating mRNA stability, while ablation of YTHDF2 results in male subfertility likely by affecting spermatid development[37,38]. However, there are no reports about how m6A-modified RNAs are regulated during meiotic metaphase or about whether an m6A reader is involved.

Here, we show that PRRC2A is highly expressed in testes and is essential for male fertility. PRRC2A-null spermatocytes show defective XY synapsis and MSCI at late prophase. PRRC2A deficiency leads to delayed metaphase entry, chromosome misalignment, spindle abnormalities, and finally arrests at metaphase I. Combining RNA-seq, Ribo-seq, and PRRC2A RIP-seq, we found that PRRC2A recognizes and downregulates spermatogonia-specific transcripts during meiosis prophase to promote the transcriptome transition from spermatogonia to spermatocytes. PRRC2A binds transcripts of genes involved in meiotic cell division and enhances their translation efficiency. Further, we found that PRRC2A interacts with YBX1, YBX2, PABPC1, FXR1, and EIF4G3, and potentially regulates mRNA metabolism and translation together with these cofactors. Our study reveals the post-transcriptional functions of PRRC2A and its essential role in the completion of meiosis I during spermatogenesis.

## Results

### PRRC2A is essential for male fertility
We found that *Prrc2a* was highly expressed in the adult testes (Fig. 1a). Immunoblotting showed that the PRRC2A protein was broadly expressed during the first wave of spermatogenesis, and we observed a rapid increase in the PRRC2A level starting from postnatal day 14 and continuing to day 22 (P14-22) (Fig. 1b), the period when early pachytene spermatocytes appear and develop through meiosis I and meiosis II to produce early round spermatids[39]. RNA in situ hybridization showed that PRRC2A mRNA was expressed at low levels in spermatogonia, spermatocytes from preleptotene stage to early/mid-pachytene stage, and step 7-10 spermatids, and was highly expressed in spermatocytes from late-pachytene stage to metaphase stage and step 1–6 spermatids (Supplementary Fig. 1a). Similarly, immunostaining

showed that the PRRC2A protein was localized in the cytoplasm of all types of germ cells except elongated spermatids, and had stronger signals and a granular distribution in both late-pachytene spermatocytes and round spermatids (Fig. 1c). In the testis, a type of germ cell-specific RNA granule known as chromatoid body (marked by DDX4 and MIWI) exists in late pachytene spermatocytes up until the round spermatid stage[40]. Co-staining of PRRC2A with DDX4 and MIWI confirmed that PRRC2A colocalizes with chromatoid bodies (Fig. 1c, Supplementary Fig. 1b).

To investigate the function of PRRC2A in spermatogenesis, we generated a mouse line with germ cell-specific knockout of *Prrc2a* (Supplementary Fig. 2a). Briefly, CRISPR-Cas9 technology was used to place loxP sites on both sides of the region from exon 2 to exon 5 in the mouse *Prrc2a* locus to generate *Prrc2a*^fl/fl mice. Then, *Stra8*-Cre transgenic mice were crossed with *Prrc2a*^fl/fl mice to generate *Prrc2a*-cko (*Stra8*-Cre; *Prrc2a*^fl/fl or *Stra8*-Cre; *Prrc2a*^fl/Δ) mice, from which *Prrc2a* was knocked out starting from differentiated spermatogonia and persisting through the later stages of the male germline (Supplementary Fig. 2a, b). After confirming the knockout of *Prrc2a* by qPCR and immunoblotting in P60 *Prrc2a*-cko testes (Supplementary Fig. 2c, d), we found that *Prrc2a*-cko adult male mice were infertile with a 37% reduction in testis weight compared to littermate controls (Fig. 1d, e). They mated with female mice (which formed normal vaginal plugs), but no offspring were produced (Supplementary Fig. 2e).

In *Prrc2a*-cko testes, the number of round spermatids decreased dramatically, and no elongating spermatids or elongated spermatids were detected (Fig. 1f). There were round spermatids that appeared with abnormal nuclear morphology and detached from the seminiferous epithelium (Fig. 1f). In the *Prrc2a*-cko epididymis, lumens did not contain sperms or had defective round spermatids (Fig. 1f, Supplementary Fig. 3b). We next performed co-staining of the acrosome marker PNA (peanut agglutinin) and DDX4 in testes from P18 to P24 mice to investigate the development of round spermatids (Supplementary Fig. 3a). At P18, there were no spermatids in the control or *Prrc2a*-cko testis. From P20 to P24, round spermatids were generated in large quantities and developed to steps 7–9 in control testes. In contrast, only a few round spermatids were produced in *Prrc2a*-cko testes, and their development did not advance beyond steps 4–6. Additionally, we found many multinucleated germ cells which appeared like incompletely divided spermatocytes or as unseparated round spermatids in P60 *Prrc2a*-cko testes and epididymides (Supplementary Fig. 3b). Most multinucleated cells contained two nuclei, and a few contained three or more nuclei, phenotypes indicative of impaired meiotic cell division[41] (Supplementary Fig. 3c).

We then performed TUNEL staining and found increased apoptosis in *Prrc2a*-cko testes, specifically noting that most of the apoptotic cells were metaphase spermatocytes or round spermatids (Fig. 1g, Supplementary Fig. 4a). Further, we staged seminiferous tubules based on the morphology and composition of germ cells and pattern of γH2AX staining (Supplementary Fig. 3d) and found that the number of apoptotic cells was greatly increased in stage I–III tubules (which contain newly produced round spermatids) (Fig. 1h). These results established that many round spermatids undergo apoptosis immediately after being generated in the *Prrc2a*-cko testes. Further, we examined apoptosis in P18 to P28 testes: compared with control testis, the number of apoptotic cells in *Prrc2a*-cko testes was similar at P18 but then increased significantly from P20 to P28 and kept at P60 (Fig. 1i). Meiotic cell division occurs around P18 to P20 during the first wave of spermatogenesis, indicating defects in *Prrc2a*-cko metaphase spermatocytes.

### PRRC2A modulates XY synapsis and MSCI
To investigate apoptosis in metaphase I spermatocytes, we first examined DNA double-strand breaks (DSBs) and homologous chromosome synapsis by staining γH2AX and SYCP3 on chromosome

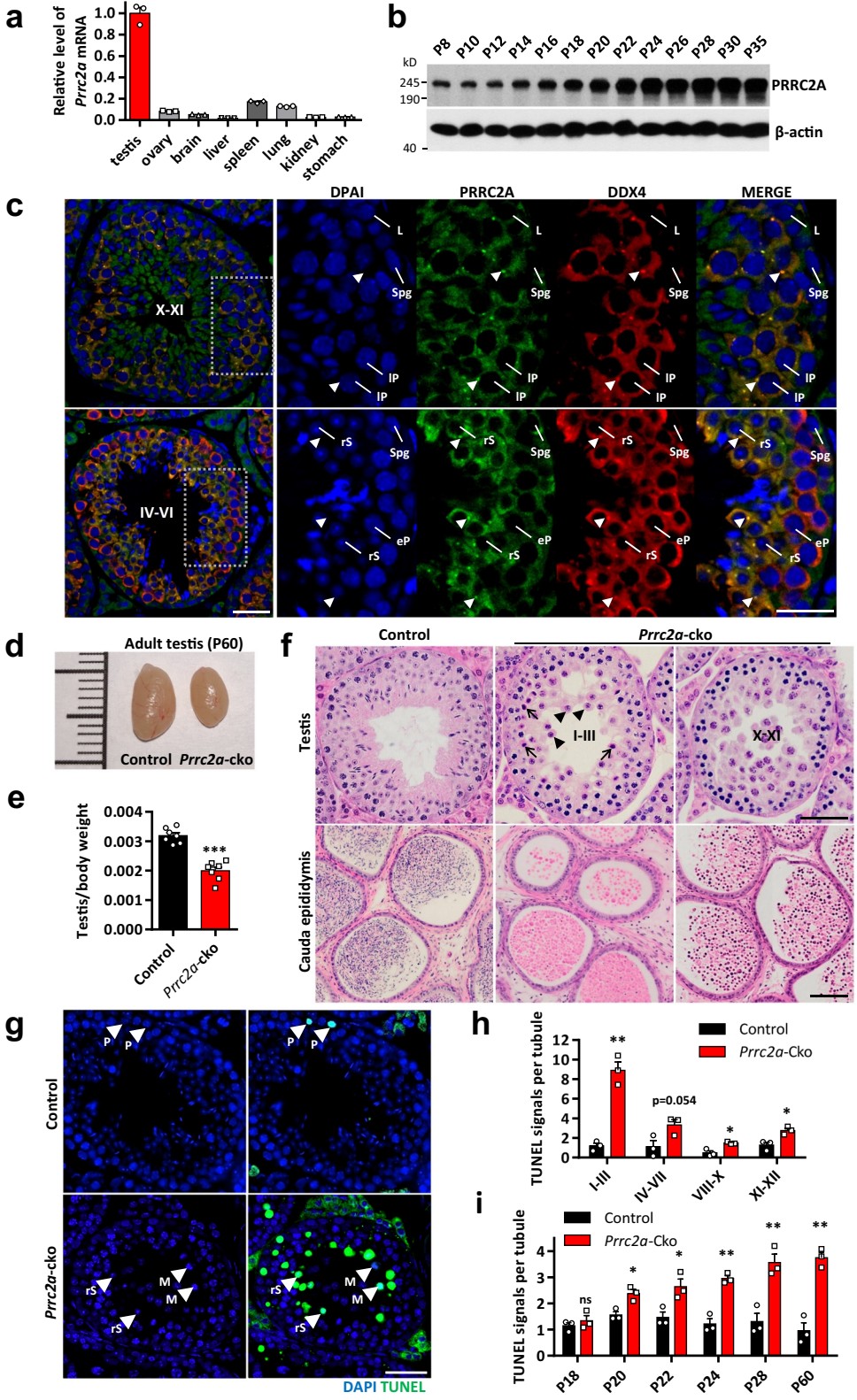

spreads of meiotic prophase spermatocytes (Fig. 2a). PRRC2A-null prophase spermatocytes develop normally through the leptotene, zygotene, pachytene, diplotene stages, and the proportion of each type of spermatocytes was similar to controls (Supplementary Fig. 4b). γH2AX signals in PRRC2A-null spermatocytes were distributed around all chromosomes at the leptotene and zygotene stages, then disappeared from autosomes but remained in XY body at the pachytene and diplotene stages, outcomes similar to the control spermatocytes

(Fig. 2a). SYCP1 and SYCP3 are essential components of the synaptonemal complex, we examined their expression on chromosomes and found autosomes in PRRC2A-null spermatocytes synapsed at the pachytene stage and separated at the diplotene stage as controls (Fig. 2a, Supplementary Fig. 4c). However, there were dramatically more pachytene spermatocytes with XY asynapsis in *Prrc2a*-cko testes (for early pachytene, control: 5.4% vs *Prrc2a*-cko: 22.0%; for late-pachytene, control: 5.9% vs *Prrc2a*-cko: 16.4%) (Fig. 2b, c). During the

**Fig. 1 | PRRC2A is highly expressed in testes and is essential for spermato-genesis. a** qPCR analysis of *Prrc2a* mRNA levels in various organs of adult mice. Two-sided student's *t*-test. Error bars, *n* = 3 mice, mean ± SEM. Source data are provided as a Source Data file. **b** WB analysis of PRRC2A protein levels in mice testes of indicated ages. **c** Immunostaining of PRRC2A and DDX4 in P60 testis sections. The right panels show enlarged images of indicated areas. Spg spermatogonia, L leptotene spermatocyte, eP early pachytene spermatocyte, lP late-pachytene spermatocyte, rS round spermatid. Arrowheads indicate chromatoid bodies. Scale bar, 20 μm. **d** Morphology of representative testes from P60 control and *Prrc2a*-cko mice. **e** Ratios of testis weight to body weight of P60 control and *Prrc2a*-cko mice (*n* = 7). Two-sided student's *t*-test. Error bars, mean ± SEM. *p* < 0.0001. Source data are provided as a Source Data file. **f** H&E staining in testis and epididymis sections of P60 control and *Prrc2a*-cko mice. Stages of the seminiferous epithelial cycle are indicated. Arrows indicate round spermatids with abnormal nuclear morphology. Arrowheads indicate detached round spermatids. Scale bar, 50 μm (up) and 100 μm (down). **g** TUNEL staining in P60 control and *Prrc2a*-cko testis sections. Arrowheads indicate germ cells with TUNEL signals. M, metaphase spermatocyte; rS, round spermatid. Scale bar, 50 μm. **h** Numbers of TUNEL-positive cells per tubule of the indicated stage in P60 control and *Prrc2a*-cko testes (*n* = 3). Two-sided student's *t*-test. Error bars, mean ± SEM. *p* = 0.0010 (for I–III), 0.0539 (for IV–VII), 0.0103 (for VIII–X), 0.0161 (for XI–XII). Source data are provided as a Source Data file. **i** Numbers of TUNEL-positive cells per tubule in control and *Prrc2a*-cko testes (*n* = 3) of indicated ages. Two-sided student's *t*-test. Error bars, mean ± SEM. *p* = 0.4448 (for P18), 0.0149 (for P20), 0.0308 (for P22), 0.0015 (for P24), 0.0071 (for P28), 0.0016 (for P60). Source data are provided as a Source Data file.

process of meiotic DSB repair, DMC1 and MLH1 are required for strand invasion and crossover formation, respectively[24]. Immunostaining showed that the number of DMC1 foci did not change after PRRC2A deletion (Fig. 2d, e). The number of MLH1 foci in PRRC2A-null spermatocytes (*n* = 22.52) was similar to the number in controls (*n* = 22.59). But in spermatocytes with asynapsed XY, the number was decreased (*n* = 21.56), corresponding to the XY asynapsis (Fig. 2f, g).

Note that XY asynapsis and impaired MSCI frequently co-occur in defective spermatocytes, such as those from Brdt[−/−][42] and Raptor[−/−][43] mice, we further examined the expression pattern of POL II (Fig. 2h). Compared to control spermatocytes, there were increased POL II signals in the XY region of PRRC2A-null pachytene and diplotene spermatocytes, clearly suggesting the disruption of MSCI (Fig. 2i). Moreover, consistent with the notion that defective MSCI causes mid-pachytene arrest and apoptosis[44,45], we found PRRC2A deficiency caused noticeably more apoptosis in mid-pachytene spermatocytes compared with control (Supplementary Fig. 4a). We next examined the expression of two MSCI factors, MDC1[46] and ATR[47], and found that their expression pattern in the XY region was not affected in PRRC2A-null spermatocytes whether the sex chromosomes were synapsed or not (Fig. 2j). The impaired MSCI may be caused by the downstream epigenetic abnormalities in the XY region, such as H3K9me3[48].

Surprisingly, fluorescence-activated cell sorting (FACS) analysis of cellular composition showed the normal number of early prophase spermatocytes but noticeably fewer late prophase spermatocytes in *Prrc2a*-cko P20 testes, although the proportion of late prophase spermatocytes became normal in adult *Prrc2a*-cko testes (Supplementary Fig. 4d). These results indicate a delayed progression from zygonema to pachynema and this delay may result in increased apoptosis starting from the early pachytene stage (Supplementary Fig. 4a).

## PRRC2A deficiency causes delayed entry and arrest at meiotic metaphase

To characterize meiotic metaphase, we performed immunostaining against the metaphase cell marker phospho-Histone H3 (Thr3) (pH3) (Fig. 3a). In *Prrc2a*-cko adult (P60) testes, the total number of pH3-positive (pH3+) spermatocytes did not differ from control testes (Fig. 3b), but the number of tubules containing pH3+ spermatocytes was significantly increased (Fig. 3c). We then quantified the number of pH3+ metaphase spermatocytes in tubules at different stages to examine their distribution (Fig. 3d). In control testes, over 95% metaphase spermatocytes appeared in the XI–XII stage tubules. However, in *Prrc2a*-cko testes, a large number of metaphase spermatocytes were also detected in stage I–III tubules, and even some in stage IV–X tubules (Fig. 3a, d). Previous studies have shown that cell division is delayed by the SAC when chromosomes cannot attach to or align on the spindle properly, and these defective cells will undergo apoptosis if defects are not repaired in time[28]. Interestingly, the distribution pattern of pH3+ spermatocytes was similar to the distribution pattern of apoptotic cells in *Prrc2a*-cko testes (Fig. 1h), indicating that PRRC2A

deficiency potentially leads to impairment of meiotic cell division and accounts for the increased apoptosis in stage I–X tubules.

Moreover, the decreased number of metaphase spermatocytes in XI–XII stages tubules of *Prrc2a*-cko adult testes also indicated delayed metaphase entry (Fig. 3d). We further examined the production of metaphase spermatocytes in juvenile testes (Fig. 3b, c). At P18, few metaphase spermatocytes were observed in both control and *Prrc2a*-cko testes. For control mice, a large number of metaphase spermatocytes were produced starting from P20 (Fig. 3b). But we observed decreased numbers of metaphase spermatocytes in *Prrc2a*-cko testes from P20 to P28, showing a slower increasing trend than controls (Fig. 3b). In control testes, around 10% of the seminiferous tubules contained metaphase spermatocytes at P20, P22, and P24 (Fig. 3c). However, <8% of tubules contained metaphase spermatocytes in *Prrc2a*-cko testes from P20 to P24 (Fig. 3c). These results suggest PRRC2A-null spermatocytes exhibited delayed metaphase entry. Moreover, from P28 to adult, the percentage of tubules contained metaphase spermatocytes in *Prrc2a*-cko testes (over 10%) was more than the percentage in controls (around 8%), also suggesting the metaphase arrest after PRRC2A deficiency.

## PRRC2A facilitates chromosome alignment and spindle assembly

The defective developmental progress of metaphase spermatocytes promoted us to investigate the influence of PRRC2A on critical processes during meiotic cell division. We observed chromosome misalignment in PPRC2A-null metaphase I spermatocytes (*Prrc2a*-cko: 26.75% vs control: 4.75%) (Fig. 3e, g). To further characterize the spindle morphology and chromosome behaviors, we performed the co-staining of α-tubulin and γ-tubulin in P60 testes sections (Fig. 3f). In PRRC2A-null spermatocytes, the misaligned chromosomes localized outside of the spindle (Fig. 3f2). γ-tubulin, a ubiquitous component of microtubule organizing centers (MTOCs), accumulates at spindle poles in control spermatocytes. In contrast, γ-tubulin foci in PRRC2A-null spermatocytes were scattered outside the spindle pole (*Prrc2a*-cko: 28.84% vs control: 0.94%) (Fig. 3f3, g) or were disintegrated at the spindle pole, phenotypes indicating the disintegrated MTOC (*Prrc2a*-cko: 11.09% vs control: 2.22%) (Fig. 3f4, g). Moreover, the fluorescence intensity of γ-tubulin at spindle poles in PRRC2A-null spermatocytes was reduced to 67% of that in control spermatocytes, suggesting defects in MTOC formation (Fig. 3h).

Given that MTOCs are known to function in spindle assembly[49], the spindle morphology possibly was affected by PRRC2A deficiency. As expected, we observed disorganized spindles in some PRRC2A-null metaphase I spermatocytes (*Prrc2a*-cko: 10.87% vs control: 0.25%), and the spindle was asymmetric with more than two spindle poles (Fig. 3f5, g). We next measured the length and width of alignment and spindles and observed no disruption of normal alignment length and spindle length in PRRC2A-null spermatocytes, but the alignment width and spindle width were significantly increased in PRRC2A-null spermatocytes (Fig. 3i, j). Additionally, the intersection angle between the

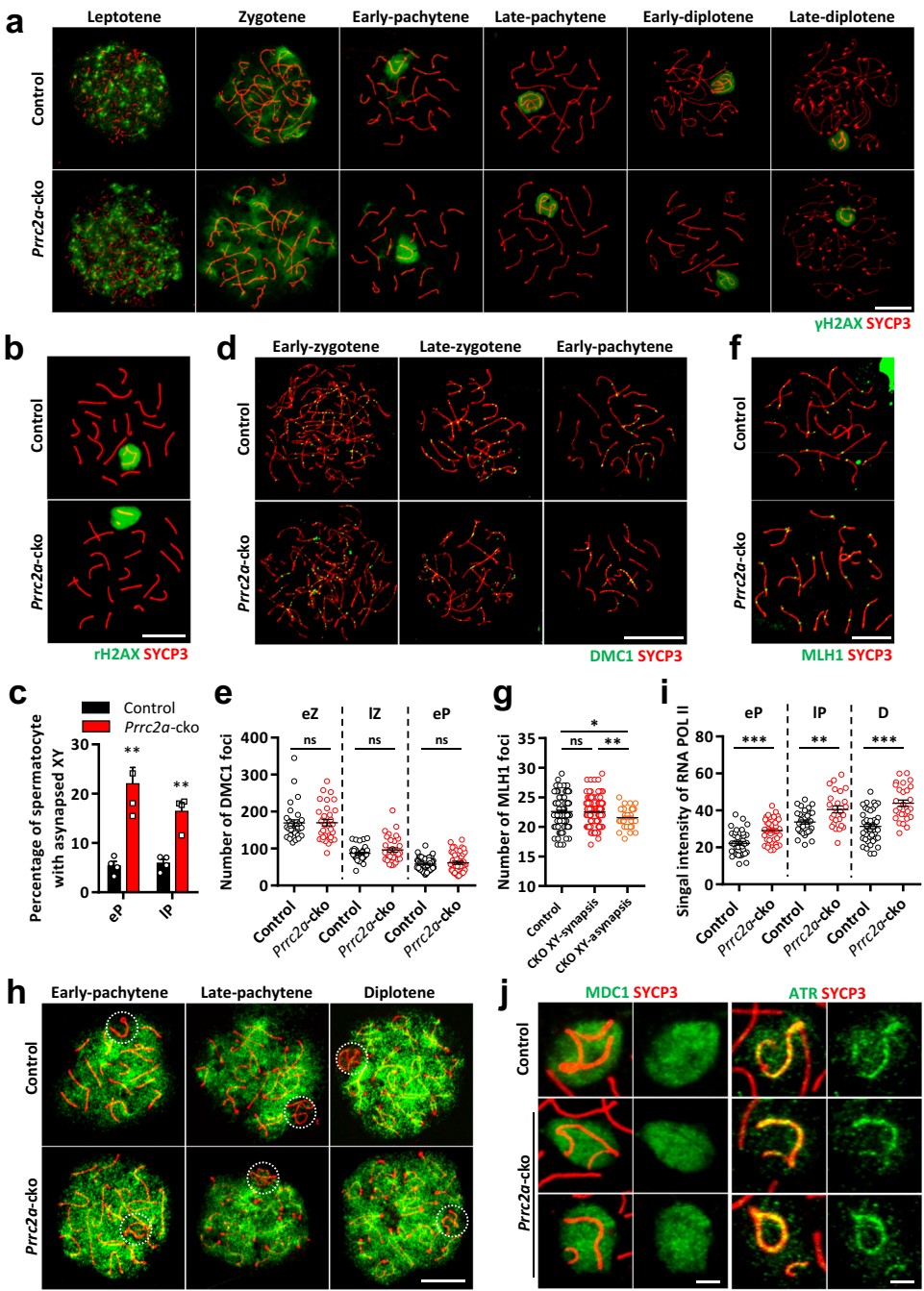

**Fig. 2 | PRRC2A deficiency leads to XY asynapsis and impaired MSCI.**
**a**, **b** Immunostaining of γH2AX and SYCP3 on chromosome spreads of control and PRRC2A-null spermatocytes. **b** Represent PRRC2A-null spermatocytes with XY asynapsis. Scale bar, 10 μm. **c** Percentage of early or late spermatocytes with XY asynapsis in total spermatocytes from P60 control and *Prrc2a*-cko testes. More than 500 chromosome spreads of spermatocytes from 4 mice were counted in each group of control and *Prrc2a*-cko. Two-sided student's *t*-test. Error bars, mean ± SEM. *p* = 0.0028 (for eP), 0.0011 (for lP). Source data are provided as a Source Data file. **d**, **e** Immunostaining of DMC1 and SYCP3 on chromosome spreads and quantification of DMC1 foci distributed on chromosomes in control and *Prrc2a*-cko early-zygotene (Control, *n* = 31; *Prrc2a*-cko, *n* = 32), late-zygotene (Control, *n* = 38; *Prrc2a*-cko, *n* = 34), and early pachytene (Control, *n* = 57; *Prrc2a*-cko, *n* = 48) spermatocytes. Two-sided student's *t*-test. Error bars, mean ± SEM. *p* = 0.9496 (for eZ), 0.1636 (for lZ), 0.4683 (for eP). Scale bar, 10 μm. Source data are provided as a Source Data file. **f**, **g** Immunostaining of MLH1 and SYCP3 on chromosome spreads

and quantification of MLH1 foci distributed on chromosomes in control (*n* = 111 cells from three mice) and *Prrc2a*-cko pachytene spermatocytes with synapsed XY-synapsis (CKO XY-synapsis, *n* = 166 cells from three mice) or asynapsed XY (CKO XY-asynapsis, *n* = 41 cells from three mice). Two-sided student's *t*-test. Error bars, mean ± SEM. ns *p* = 0.7998, **p* = 0.0147, ***p* = 0.0065. Scale bar, 10 μm. Source data are provided as a Source Data file. **h** Immunostaining of POL II and SYCP3 on chromosome spreads of control and PRRC2A-null spermatocytes. Circles indicate XY bodies. Scale bar, 10 μm. **i** Quantification of signal intensity of POL II within XY body of control and PRRC2A-null spermatocytes. eP early pachytene spermatocyte (*n* = 39, 39); lP late-pachytene spermatocyte (*n* = 31, 26); D diplotene spermatocyte (*n* = 46, 28). Two-sided student's *t*-test. Error bars, mean ± SEM. ***p* = 0.0021, ****p* < 0.0001. Source data are provided as a Source Data file. **j** XY regions of control and *Prrc2a*-cko pachytene spermatocytes immunostained with silencing factors (MDC1 or ATR) and SYCP3. Scale bar, 2 μm.

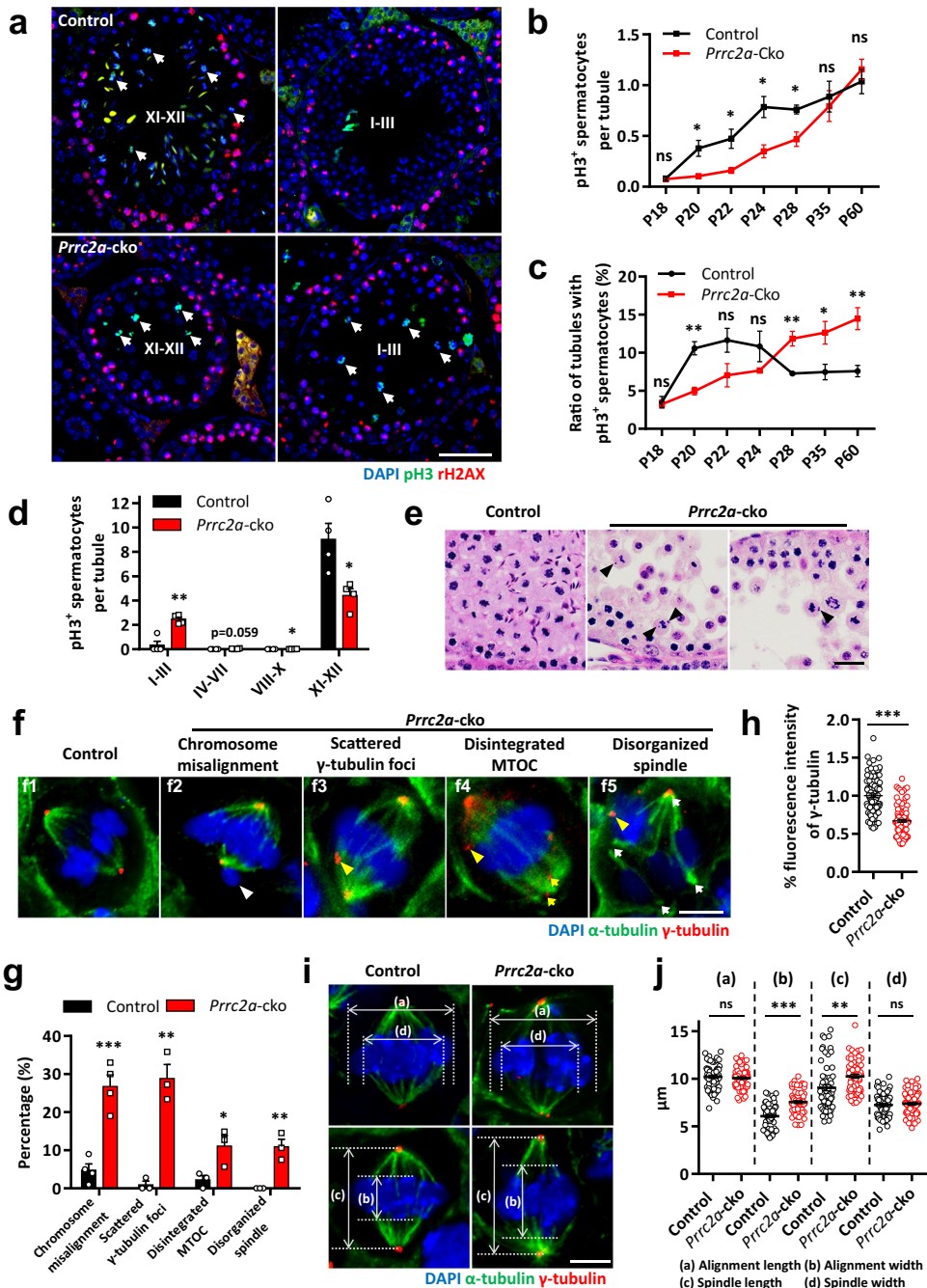

spindle polarity axis and the equatorial plate did not differ from the control cells (Supplementary Fig. 4e, f). Thus, PRRC2A is required for proper chromosome alignment and spindle morphology, and PRRC2A deficiency most likely activated the spindle assembly checkpoint and caused metaphase I arrest.

Past studies suggested the lack of kinetochore-microtubule attachment or impaired centrosome assembly leads to defective alignment and spindle[27,28,49]. So, we performed the co-staining of CREST (a marker of kinetochore) and α-tubulin and found normal kinetochore-microtubule attachment in PRRC2A-null spermatocytes, even in the case of chromosome misalignment and spindle disorganization (Fig. 4a). To examine centrosome assembly, we examined the expression of CEP192 (a protein involved in centrosome assembly) and found that the signal intensity of CEP192 at the spindle pole was significantly reduced−or even disappeared−in PRRC2A-null spermatocytes (Fig. 4b, e). Previous studies have shown that CEP192 is

required for spindle formation in oocytes[50] and CEP192 deficiency in Hela cells leads to malformed spindles, abnormal chromosomal alignment, and formation of multinucleated cells[51,52], findings similar to our results (Supplementary Fig. 3b, c). The immunoblotting also showed the decreased protein abundance of CEP192 and CEP152 (a centrosome protein required for the organization of MTOC in meiosis of mouse oocytes[50]) in PRRC2A-null spermatocytes (Figs. 4c, d and 7e), indicating the centrosome defects.

Additionally, we examined the metaphase-promoting factor (MPF), which is required for prophase-to-metaphase transition in both mitosis and meiosis[53]. MPF is composed of cyclin-dependent kinase 1 (CDK1) and cyclin B1 (CCNB1). We found decreased protein abundance of CDK1 which was consistent with the observation of metaphase entry delay, although the level of CCNB1 was normal (Fig. 4c, d). We also examined the expression of mitotic marker CCNA2[54] and found its protein abundance was not affected in PRRC2A-null spermatocytes (Fig. 4c, d).

**Fig. 3 | PRRC2A deficiency leads to chromosome misalignment and spindle disorganization. a** Immunostaining of pH3 and γH2AX in P60 control and *Prrc2a*-cko testis sections. Seminiferous tubule stages are indicated. Arrowheads indicate metaphase spermatocytes. Scale bar, 50 μm. **b** The number of pH3⁺ spermatocytes per seminiferous tubule in control and *Prrc2a*-cko testes of indicated ages (n = 4 mice for P18, P20, P60; n = 3 mice for P22, P24, P28, P35). Two-sided student's *t*-test. Error bars, mean ± SEM. p = 0.7785 (for P18), 0.0149 (for P20), 0.0334 (for P22), 0.0231 (for P24), 0.0250 (for P28), 0.6855 (for P35), 0.4705 (for P60). Source data are provided as a Source Data file. **c** The ratio of seminiferous tubules with pH3⁺ spermatocytes to total tubules in control and *Prrc2a*-cko testes of indicated ages (n = 4 mice for P18, P20, P60; n = 3 mice for P22, P24, P28, P35). Two-sided student's *t*-test. Error bars, mean ± SEM. p = 0.7580 (for P18), 0.0014 (for P20), 0.1047 (for P22), 0.1941 (for P24), 0.0092 (for P28), 0.0456 (for P35), 0.0054 (for P60). Source data are provided as a Source Data file. **d** The number of pH3⁺ spermatocytes per seminiferous tubules of the indicated stage in P60 control and *Prrc2a*-cko testes. Two-sided student's *t*-test. Error bars, n = 4 mice, mean ± SEM. p = 0.0014 (for I–III), 0.0591 (for IV–VII), 0.0242 (for VIII–X), 0.0176 (for XI–XII). Source data are provided as a Source Data file. **e** H&E staining in P60 control and *Prrc2a*-cko testes sections. Arrowheads indicate misaligned chromosomes in metaphase I spermatocytes. Scale bar, 20 μm. **f** Representative metaphase I spermatocytes in

P60 control and *Prrc2a*-cko testis sections immunostained with α-tubulin and γ-tubulin. White arrowheads indicate misaligned chromosomes, yellow arrowheads indicate scattered γ-tubulin foci outside the spindle pole, yellow arrows indicate disintegrated MTOC at the spindle pole, white arrows indicate spindle poles. Scale bar, 5 μm. **g** Percentage of metaphase I spermatocytes with chromosomes misalignment (n = 4 biological independent mice), scattered γ-tubulin foci, disintegrated MTOC, and disorganized spindle (n = 3 biological independent mice) in P60 control and *Prrc2a*-cko testes. Two-sided student's *t*-test. Error bars, mean ± SEM. p = 0.0007, 0.0018, 0.0423, 0.0057. Source data are provided as a Source Data file. **h** Fluorescence intensity of γ-tubulin at the spindle pole relative to that in control (n = 97, 100 cells). Two-sided student's *t*-test. Error bars, mean ± SEM. ***p < 0.0001. Source data are provided as a Source Data file. **i, j** The average length and width of the alignment and spindles in control and *Prrc2a*-cko metaphase I spermatocytes immunostained with α-tubulin and γ-tubulin (n = 61 cells for control, n = 64 cells for *Prrc2a*-cko). Two-sided student's *t*-test. Error bars, mean ± SEM. The value of each data is 10.21 ± 0.16, 10.07 ± 0.13, 6.07 ± 0.15, 7.55 ± 0.15, 9.05 ± 0.32, 10.25 ± 0.22, 7.25 ± 0.16, 7.38 ± 0.15 from left to right. p = 0.5051 (ns), <0.0001 (***), 0.0018 (**), 0.5759 (ns). Scale bar, 5 μm. Source data are provided as a Source Data file.

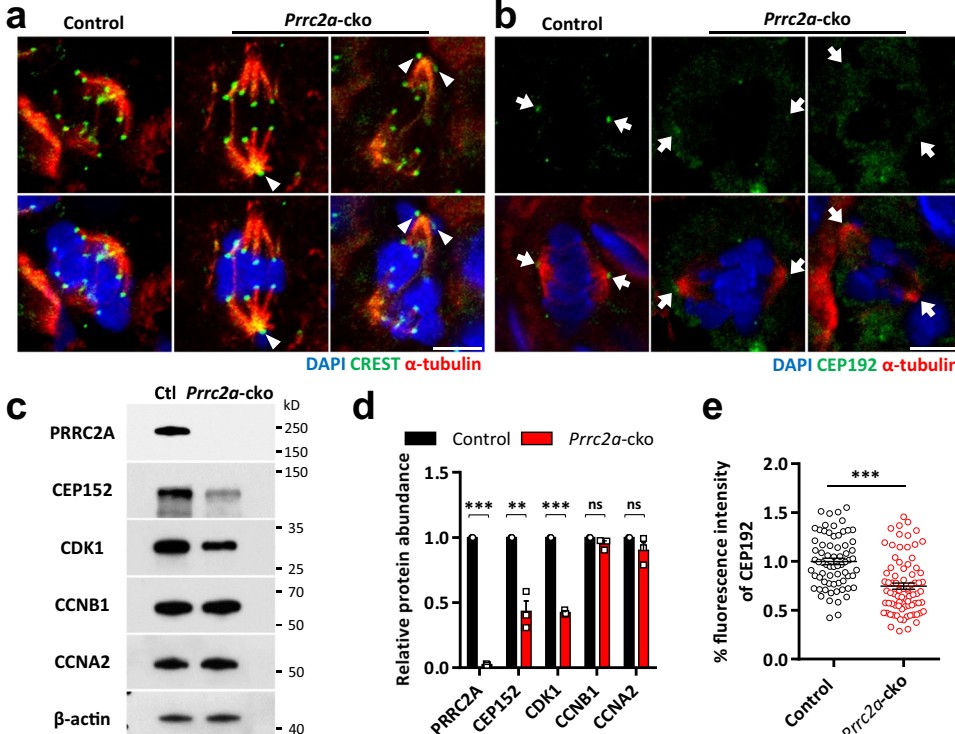

**Fig. 4 | PRRC2A deficiency leads to impaired centrosome and downregulation of MPF. a** Representative metaphase I spermatocytes in P60 control and *Prrc2a*-cko testes sections immunostained with CREST and α-tubulin. Scale bar, 5 μm. **b** Representative metaphase I spermatocytes in P60 control and *Prrc2a*-cko testes sections immunostained with CEP192, α-tubulin. Arrows indicate spindle poles. Scale bar, 5 μm. **c** WB test of indicated proteins in control and PRRC2A-null spermatocytes from mice. **d** Quantification of the relative level of indicated proteins in

control and PRRC2A-null spermatocytes. Two-sided student's *t*-test. Error bars, n = 3 mice, mean ± SEM. ns p = 0.1088 (for CCNB1), 0.1735 (for CCNA2), **p = 0.0022, ***p < 0.0001. Source data are provided as a Source Data file. **e** Quantification of fluorescence intensity of CEP192 foci at spindle pole (n = 69, 79 cells). Two-sided student's *t*-test. Error bars, mean ± SEM. ***p < 0.0001. Source data are provided as a Source Data file.

## PRRC2A reduces abundance and promotes translation of its targets

The majority of cytoplasmic m6A readers have been shown to regulate m6A-modified transcripts by affecting their mRNA stability and/or their translation efficiency[1]. To explore the mechanism(s) through which PRRC2A regulates spermatogenesis, we purified Hoechst 33342 stained prophase spermatocytes (from leptotene to diplotene) from adult *Prrc2a*-cko and control testes using FACS according to previously reported method[55] (Supplementary Fig. 4d), then assessed

their transcriptome and translatome There were 2033 differentially transcribed genes (DTG, up: 1140 and down: 893) and 686 differential translation efficiency genes (DTEG, up: 287 and down: 399) (Fig. 5a, Supplementary Data 1). Among them, the mRNA abundance and translation efficiency of 476 genes were both differentially changed (DTG&DTEG), 1557 genes were exclusive DTGs, and 210 genes were exclusive DTEGs (Fig. 5a). Gene ontology (GO) analysis indicated that the downregulated DTGs were enriched for annotated functions related to spermatogenesis and cilium movement, but no enrichment for

spermatogenesis-related terms was found among the upregulated DTGs (Fig. 5b). Conversely, terms related to male production were enriched among upregulated DTEGs, and terms enriched among downregulated DTEGs were not related to spermatogenesis (Fig. 5c). Thus, transcriptome and translational efficiency profile were altered by PRRC2A deficiency and showed the opposite change patterns.

Additionally, the majority of X and Y chromosome-linked genes were upregulated in PRRC2A-null spermatocytes (168 up vs. 8 down) (Supplementary Fig. 6a), a trend corresponding to the impaired MSCI we observed in PRRC2A-null spermatocytes (Fig. 2h, i). We also confirmed the upregulation of several representative genes by qPCR (Supplementary Fig. 6b).

To identify the targets of PRRC2A, we inserted a 3×flag tag after the *Prrc2a* CDS region to generate *Prrc2a*-flag transgenic mice by CRISPR-Cas9 (Supplementary Fig. 5a, b) and tested whether these mice were suitable for the subsequent PRRC2A-RIP experiment. We found that the PRRC2A-Flag proteins were expressed in transgenic testes with similar abundance to controls and were efficiently pulled down using the anti-Flag antibody (Supplementary Fig. 5c). The immunofluorescence co-staining of Flag, PRRC2A, and DDX4 showed that PRRC2A-Flag proteins had the same expression pattern as PRRC2A (Fig. 1c, Supplementary Fig. 5d). These results supported the use of this transgenic mouse line for subsequent immunoprecipitation (IP) experiments.

We then performed PRRC2A RIP-seq using P20 *Prrc2a*-flag mouse testes to identify candidate binding target RNAs of PRRC2A. A total of 9442 peaks from 5,981 genes were identified in two biological replicates (Supplementary Data 2), most of which (94.31% and 96.42% for each replicate) were for mRNAs (Fig. 5d and Supplementary Fig. 6c). PRRC2A-binding peaks were enriched in the CDS and 3'UTR region of mRNAs and were highly enriched near the stop codon (Fig. 5d, e), results consistent with the previously reported distribution patterns for m6A modification generally[56,57] and for PRRC2A-binding peaks specifically[21]. A motif analysis identified the known consensus m6A motif GGACU[56,57] among PRRC2A-binding peaks (Fig. 5f). Further, we overlapped our PRRC2A-binding peak dataset with m6A-modified peaks from a previously reported P20 MeRIP-seq dataset[11] and found that 3366 peaks (from 2263 transcripts) of 9442 (35.65%) PRRC2A-binding peaks carried m6A modifications (Supplementary Fig. 6d, Supplementary Data 3). These findings support that PRRC2A functions as an m6A reader in testes.

We then followed PRRC2A-binding transcripts and reported m6A-modified sites in the RNA-seq and Ribo-seq data and 4081 PRRC2A-binding targets (1954 of them contain m6A modifications) were obtained (Fig. 5g). PRRC2A targets showed more increased RNA abundance compared with non-targets in PRRC2A-null spermatocytes (Fig. 5h). Conversely, translational efficiency (TE) of PRRC2A targets became lower than non-targets in PRRC2A-null spermatocytes (Fig. 5h). Remarkably, as the number of PRRC2A-binding m6A-modified peaks increased, the RNA abundance increased and the TE decreased correspondingly, and consequently, transcripts with more than three binding peaks exhibited the highest RNA abundance and the lowest TE (Fig. 5i). Therefore, PRRC2A reduces the abundance of its targets and improves their translation efficiency during the male meiotic prophase. Moreover, PRRC2A-binding methylated transcripts showed higher RNA abundance and lower TE than PRRC2A-binding targets without m6A modifications, this result indicated that PRRC2A exerts functions dependently on m6A and confirmed its role as an m6A reader (Fig. 5i).

### PRRC2A promotes transcriptome transition from spermatogonia to spermatocytes

To better understand the expression changes in PRRC2A-null spermatocytes, we compared our RNA-seq and Ribo-seq data with published transcriptomes of sorted spermatogenic cell types[58]. Genes with increased RNA and ribosome-protected fragment (RPF) abundance were preferentially expressed in spermatogonia, whereas the downregulated genes were preferentially expressed in spermatocytes and spermatids (Fig. 6a). Correspondingly, we found that spermatogonia-specific genes were upregulated with increased RNA and RPF abundance in PRRC2A-null spermatocytes (Fig. 6b, c), including genes involved in spermatogonial stem cell maintenance (*Id4*, *Gfra1*, *Ret*, *Nanos3*, *Plzf*, *Sall4*, *Foxo1*, *Etv5*) and spermatogonial differentiation (*Stra8*, *Kit*, *Dmrt1*, *Sox3*) (Fig. 6e). However, spermatocyte-specific genes (such as *Ccdc36* and *Aurkc*) and spermatid-specific genes (including several genes essential for spermiogenesis, such as *Prm1/2/3* and *Tnp1/2*) were downregulated in PRRC2A-null spermatocytes (Fig. 6b, c, e). The gene set enrichment analysis (GSEA) of RNA-seq and Ribo-seq data using cell-type-specific gene sets (Supplementary Data 4) yielded similar results (Fig. 6d). Interestingly, we noticed genes required for meiotic recombination (such as *Spo11* and *Hormad1*) were not affected by PRRC2A deficiency (Fig. 6e) which explained why PRRC2A-null spermatocytes were able to progress through meiotic prophase, although with delayed progress in juvenile testes and increased apoptosis starting from pachytene stage in adult testes. These results suggested PRRC2A is required for the transition of expression profile from spermatogonia to spermatocyte and the upregulation of genes required for spermiogenesis. PRRC2A deficiency caused partial loss of spermatocyte identity, and consequently leads to multiple meiotic defects and failed spermiogenesis.

Further, the majority of PRRC2A-binding methylated targets were upregulated and overlapped more with spermatogonia-specific genes than other cell-type-specific genes (Fig. 5h, 6a, Supplementary Fig. 6e). We obtained 59 upregulated target transcripts from spermatogonia-specific gene sets, among them, *Plzf*, *Sall4*, *Foxo1*, and *Sox3* were four representative genes known to regulate spermatogonial stem cell maintenance or differentiation (Fig. 6e, Supplementary Fig. 6e). Their transcripts contained overlapped PRRC2A-binding peaks and m6A modification sites (Fig. 6f), and their increased RNA abundance was verified in PRRC2A-null spermatocytes by qPCR (Fig. 6g). Thus, PRRC2A recognizes spermatogonia-specific transcripts and downregulates their expression during meiotic prophase to promote the transition from mitosis to meiosis. Moreover, although alterations of RPF abundance in PRRC2A-null spermatocytes showed a pattern similar to that of RNA abundance, the TE profile had an opposite alteration pattern (Fig. 6a), possibly suggesting a type of translational balance mechanism in spermatocytes to compensate for the disruption of the transcriptional landscape (but failed).

### PRRC2A promotes translation of genes involved in meiotic cell division

To investigate the potential regulatory impacts of PRRC2A during meiotic metaphase, we examined the expression of genes involved in meiotic cell division by GSEA (using the GO database) in RNA-seq and Ribo-seq data. We found that gene sets annotated with "meiotic cell cycle", "meiotic nuclear division", and "spindle" showed increased RNA abundance but decreased RPF in PRRC2A-null spermatocytes (Fig. 7a), which indicated that PRRC2A deficiency led to reduced translation efficiency of these genes. Previous studies reported that *Cep192*[50], and *Wnk1*[59] regulate cell division, and their deficiency leads to impaired spindle assembly and chromosome alignment during meiosis or mitosis, phenotypes similar to the defects we observed in PRRC2A-null metaphase spermatocytes (Fig. 3f). RNA immunoprecipitation data showed that their transcripts bore overlapped PRRC2A-binding peaks and m6A modification sites (Fig. 7b), and we also verified that using PRRC2A RIP-qPCR and MeRIP-qPCR (Fig. 7c, d). We found that protein levels of CEP192 and WNK1 were reduced in PRRC2A-null spermatocytes (Fig. 7e, f), but their RNA abundance was not affected or slightly increased after PRRC2A deficiency (Fig. 7g), which indicates reduced

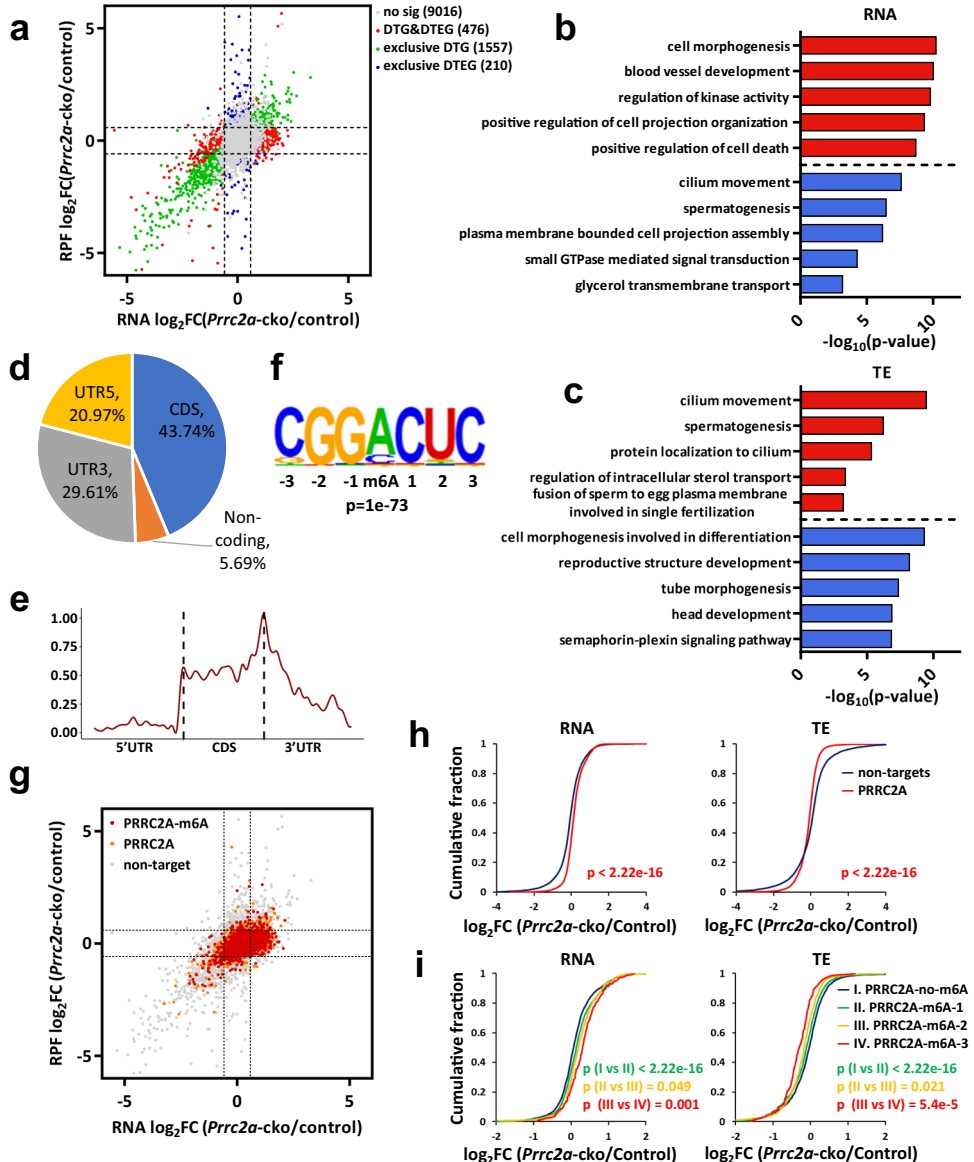

**Fig. 5 | PRRC2A decreases the mRNA abundance and improves the translation efficiency of its targets. a** Scatter plots of Ribo-seq vs RNA-seq in control and PRRC2A-null spermatocytes. Genes were classified according to their regulation with the parameters indicated criteria. Gray dots (no sig) indicate genes with no significant change. Green dots (exclusive DTG) indicate genes with significant changes in RNA abundance (fold change (FC) > 1.5, $p$-adjusted < 0.05) and with no change in translation efficiency. Blue dots (exclusive DTEG) indicate genes with significant changes (FC > 1.5, $p$-value < 0.05) in translation efficiency and with no change in RNA abundance. Red dots (DTG&DTEG) indicate genes with significant changes in RNA abundance and translation efficiency. **b, c** Top GO terms in biological process categories of down and upregulated DTGs (**b**) and DTEGs (**c**). **d** The pie chart shows the distribution region of PRRC2A-binding peaks on transcripts in one of two repeats. **e** Distribution of PRRC2A-binding peaks along with transcripts.

**f** Consensus binding motif of PRRC2A identified by HOMER ($p = 1e^{-73}$). **g** Scatter plots of Ribo-seq vs RNA-seq in control and PRRC2A-null spermatocytes. Genes were classified into groups according to indicated criteria. Gray dots (non-target) indicate genes with no PRRC2A-binding sites. Orange dots indicate genes with PRRC2A-binding sites. Red dots indicate genes with overlapped PRRC2A-binding and m6A-modified sites. **h, i** Cumulative distribution of RNA abundance (left two panels) and translational efficiency (right two panels) changes between control and PRRC2A-null spermatocytes. Top two panels show non-targets (blue), PRRC2A-RIP targets (red) (H). Bottom two panels show targets with PRRC2A-binding m6A-not-modified sites (blue) and targets with one (green), two (orange), and more than three (red) PRRC2A-binding m6A-modified sites (I). $p$-values were calculated using two-sided Wilcoxon test.

translational efficiency of these transcripts in PRRC2A-null spermatocytes. Additionally, transcripts of *Dazl* contained no m6A modification or PRRC2A-binding peak (Fig. 7b–d) and were used as the control. Its mRNA level and protein level were not changed in PRRC2A-null spermatocytes (Fig. 7e–g). Therefore, PRRC2A recognizes transcripts of genes involved in meiotic cell division and promotes the translation, thereby facilitating the progression of male meiotic metaphase.

To explore the mechanism of PRRC2A-mediated regulation of mRNA metabolism and translation, we further performed co-IP

coupled with mass spectrometry (IP-MS) for PRRC2A and identified a series of interacting proteins (Supplementary Fig. 7a, b). GO analysis of the candidate interacting proteins showed enrichment for functional annotations related to mRNA metabolism and cell cycle (Supplementary Fig. 7c).

Among them, Y-Box Binding Protein 1 and 2 (YBX1 and YBX2), and fragile-X mental retardation autosomal 1 (FXR1) were verified to bind PRRC2A by co-IP (Fig. 7h, Supplementary Fig. 7b), and previous studies showed that they could promote mRNA degradation[60-62]. We also

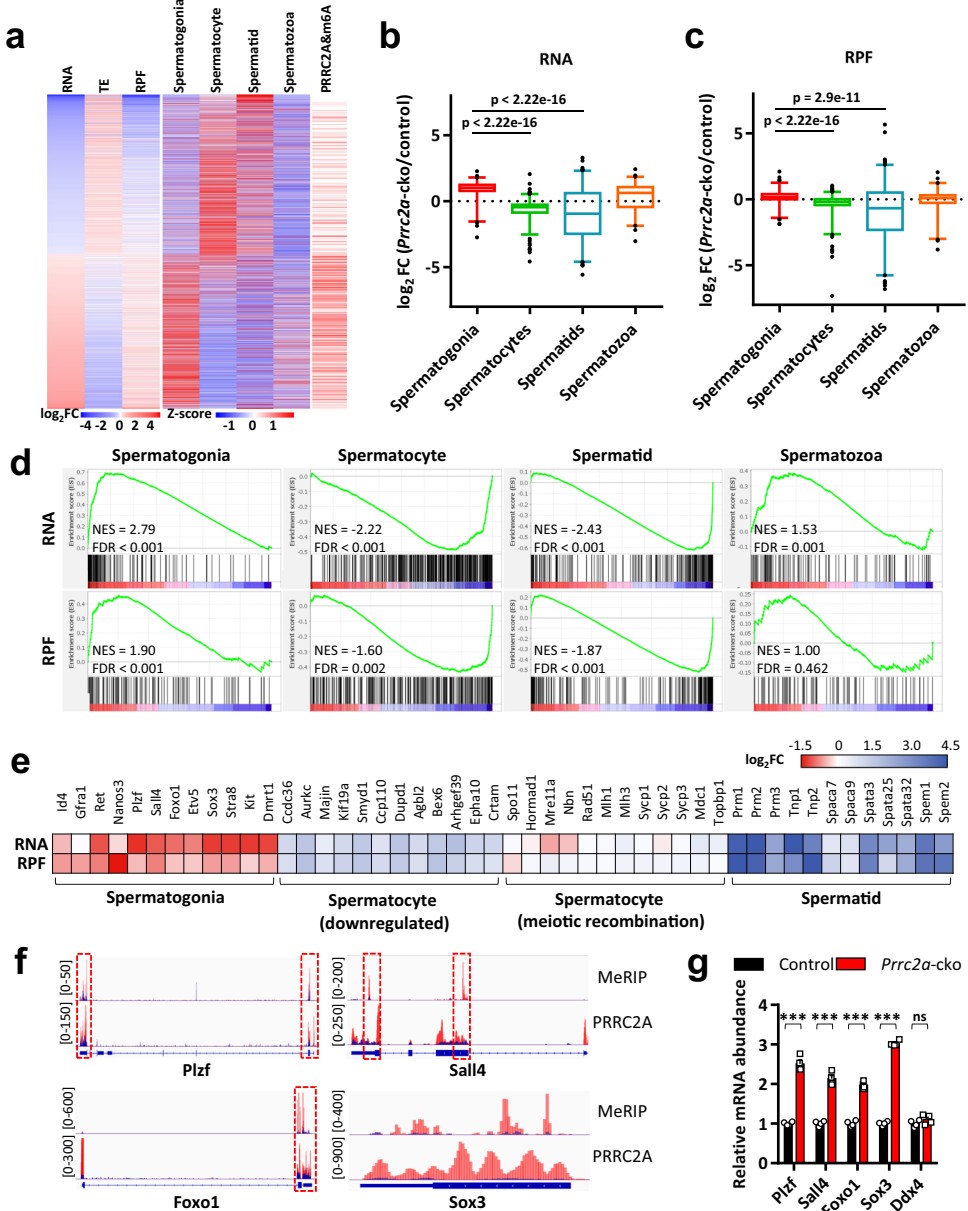

**Fig. 6 | PRRC2A deficiency causes defective transcriptome transition from spermatogonia to spermatocytes. a** Heatmap showing the changes of RNA abundance, TE, and RPF for genes with differentially RNA abundance (*p*-adjusted < 0.05 as cutoff) in PRRC2A-null spermatocytes versus control spermatocytes. Genes with PRRC2A-binding m6A-modified sites and their expression levels in different cell types were correspondingly shown. **b**, **c** Box plots show the RNA and RPF abundance of cell-type-specific genes from RNA-seq and Ribo-seq data (*Prrc2a*-cko versus control, two-sided Wilcox test) (*n* = 186, 395, 267, 125 genes). The box indicates the lower (25%) and upper (75%) quantile and the white line indicates the median. Whiskers extend from 2.5% to 97.5% percentile, non-outlier data points. **d** GSEA analysis of cell-type-specific gene sets in RNA-seq and Ribo-seq data (*Prrc2a*-cko versus control). **e** A heatmap showing the fold change of RNA and RPF abundance for representative cell-type-specific genes from RNA-seq and Ribo-seq data (*Prrc2a*-cko versus control). **f** Integrative Genomics Viewer shows the distribution of m6A-modified peaks and PRRC2A-binding peaks along with indicated transcripts in PRRC2A RIP-seq and MeRIP-seq[11] data. Blue peaks and red peaks represent reads in the input and IP groups, respectively. Red boxes show the area containing both m6A-modified peaks and PRRC2A-binding peaks. **g** Relative abundance of indicated mRNAs in control and PRRC2A-null spermatocytes detected by qPCR. Two-sided student's *t*-test. Error bars, *n* = 4 samples, mean ± SEM. ns *p* = 0.2311, ***p* < 0.0001. Source data are provided as a Source Data file.

found that PRRC2A interacted with poly(A) tail binding protein C1 (PABPC1) and eukaryotic translation initiation factor 4 gamma 3 (EIF4G3) (Fig. 7h, Supplementary Fig. 7b) which were reported to facilitate translation initiation during spermatogenesis[63,64]. A recent study showed that FXR1 also promotes translation together with EIF4G3 and PABPC1 to drive spermiogenesis[65]. EIF4G3 has also been shown to be required for male fertility, and its deficiency causes failed entry to the meiotic metaphase during spermatogenesis, defects also observed in PRRC2A-null spermatocytes[64]. We found the protein abundance of EIF4G3 was decreased in PRRC2A-null spermatocytes, while the abundance of its reported downstream, HSPA2[64], was normal (Fig. 7i). Other cofactors of PRRC2A were not affected by PRRC2A deficiency (Fig. 7i). Collectively, PRRC2A potentially recruits different transcripts and corresponding cofactors to promote mRNA decay and/or translation of its targets during meiotic prophase.

Additionally, we found the interaction between PRRC2A with YBX1 and YBX2 can be disrupted by RNase treatment, while the interaction of PRRC2A with other cofactors resisted this treatment.

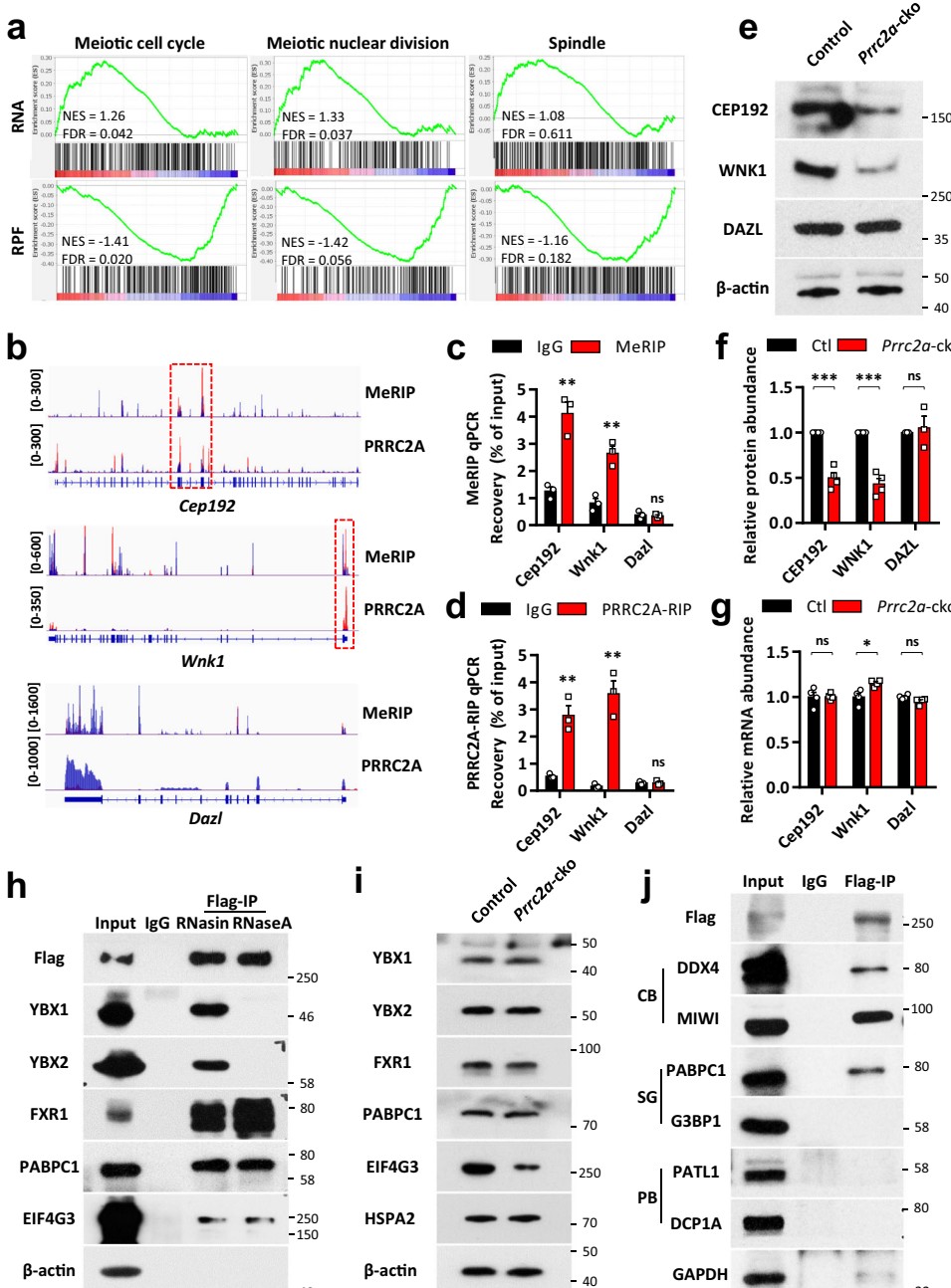

**Fig. 7 | PRRC2A promotes the expression of genes involved in meiotic cell division. a** GSEA analysis of indicated gene sets in RNA-seq and Ribo-seq data (*Prrc2a*-cko versus control). **b** Integrative Genomics Viewer showed the distribution of m6A-modified peaks and PRRC2A-binding peaks along with indicated transcripts in PRRC2A RIP-seq and m6A-seq[11] data. Blue peaks and red peaks represent reads in the input and IP groups, respectively. Red boxes show the area containing both m6A-modified peaks and PRRC2A-binding peaks. **c, d** MeRIP-qPCR and PRRC2A RIP-qPCR analysis of indicated transcripts in P20 testes. Two-sided student's *t*-test. Error bars, *n* = 3 biological repeats, mean ± SEM. ns *p* > 0.05, \*\**p* < 0.01. Source data are provided as a Source Data file. **e** WB test of indicated protein in control and PRRC2A-null spermatocytes. **f, g** Quantification of the relative protein (**f**) and RNA (**g**) level of indicated genes in control and PRRC2A-null spermatocytes. Two-sided student's *t*-test. Error bars, *n* = 3 biological repeats, mean ± SEM. ns *p* > 0.05, \**p* < 0.05, \*\*\**p* < 0.001. Source data are provided as a Source Data file. **h** Testis lysates were subjected to IP with anti-Flag or IgG control antibodies. IP groups were treated with RNase inhibitor (RNasin) or RNaseA respectively. Indicated proteins were detected by WB. **i** WB test of indicated protein in control and PRRC2A-null spermatocytes. **j** Testis lysates were subjected to IP with anti-Flag or IgG control antibodies. Indicated proteins were detected by WB. CB chromatoid body, SG stress granule, PB processing body.

So, we believed that PRRC2A indirectly binds YBX1 and YBX2 and this interaction is dependent on the existence of RNA (Fig. 7h).

Notably, both YBX2 and PABPC1 are components of the chromatoid body[66], and we also successfully verified the interactions of PRRC2A with two marker proteins of the chromatoid body, DDX4 and MIWI (Fig. 7j). Given that the chromatoid body was reported to harbor components of nonsense-mediated mRNA decay (NMD) pathway[66]

and supports NMD of long 3' UTR mRNAs[67], we speculated that PRRC2A potentially recruits transcripts into the chromatoid body and mediates their decay. Further, several previous studies have reported that cytosolic m6A readers co-localize with stress granules and/or with processing bodies[6,7,17,21,68,69]. In testes, we found no evidence that PRRC2A interacts with a stress granule marker (G3BP1), or with markers of processing bodies (PATL1, DCP1A) (Fig. 7j). However, PABPC1,

EIF4G3, and FXR1 are key components of the stress granule[70], suggesting PRRC2A may partly interact with stress granules. Previous reports showed that stress granules can interact with P bodies or undergo autophagy to mediate RNA degradation in stress conditions such as heat stress, oxidative stress, or viral infections[71]. Therefore, PRRC2A may promote mRNA degradation with stress granules in male germ cells under stress.

## Discussion

Past studies reported that N6-Methyladenosine (m6A) exists in male germ cells[29]. m6A writers such as METTL3 and METTL14[29,30], m6A erasers like ALKBH5 and FTO[5,31] are required for spermatogenesis, indicating an important role of m6A in spermatogenesis. As the functional mediator of m6A, multiple m6A readers have been functionally linked to the development of spermatogonia (YTHDF2[37] and YTHDC1[36]), meiotic initiation, pachytene progression (YTHDC2[7,8,11,32,33]), and spermiogenesis (YTHDF2[38]). But recent studies showed that YTHDC2 regulates spermatogenesis independent of m6A recognition[34,35]. These previous studies have collectively showcased the diverse regulatory influences of m6A in spermatogenesis. In the present study, we demonstrate that m6A reader PRRC2A is essential for the completion of meiosis I.

Detailly, we generated a transgenic mouse line with conditional knockout of *Prrc2a* in male germ cells and found that PRRC2A deficiency leads to male sterility. PRRC2A-null spermatocytes exhibit delayed developmental progress in juvenile testes and show defective XY synapsis and MSCI at late prophase in adult testes. Combining the results of histological analysis and immunostaining, we found that PRRC2A deficiency leads to delayed metaphase entry. PRRC2A-null spermatocytes exhibit chromosome misalignment and spindle disorganization, and consequently, are probably delayed by the SAC and undergo apoptosis. Some escape from SAC may produce round spermatids with abnormal nuclear morphology or multinucleated cells. These defective round spermatids cannot develop to the advanced stage and undergo apoptosis soon. Notably, the first wave of spermatogenesis is not the same as adult spermatogenesis in the *Prrc2a*-cko testes. Although the delayed entry and defective completion of metaphase occur during both the first wave of spermatogenesis (Figs. 1i, and 3b, c) and in adult spermatogenesis (Figs. 1h and 3d), the delayed progression from zygonema to pachynema only occur during the first wave of spermatogenesis (Supplementary Fig. 4d). The progression during the meiotic prophase of adult spermatogenesis was normal in *Prrc2a*-cko testes (Fig. 2a, Supplementary Fig. 4b).

In mammalian cells, cytosolic m6A readers generally affect mRNA metabolism and translation efficiency. YTHDF2[6] and YTHDF3[10] mediate RNA decay, while IGF2BPs[17], FMRP[19], SND1[20], and PRRC2A[21] stabilize RNA targets. YTHDF1[13], YTHDF3[9,10], IGF2BPs[17], eIF3[18] were reported to modulate mRNA translation. Here, we showed that PRRC2A is a cytosolic m6A reader in male germ cells that mediated the decay of its mRNA targets or improves the translation efficiency, which is different from the previous study[21]. During meiotic prophase, PRRC2A binds and downregulates spermatogonia-specific transcripts to promote the transcriptome transition from spermatogonia to spermatocytes. Noted that the development of differentiated spermatogonia and early prophase spermatocytes is not affected by PRRC2A deficiency, this may be because PRRC2A does not play a central role at these stages. The expression of PRRC2A is low during these stages (Fig. 1b, c, S1) and some other proteins could also promote the transition from mitosis to meiosis, such as YTHDC2[7,8,11,32,34,35]. Further, *Ccna2*, the typical gene whose overexpression results in impaired meiosis initiation[54], is expressed normally in PRRC2A-null spermatocytes (Fig. 4c, d). So, the development of differentiated spermatogonia and early prophase spermatocytes may be less affected and could still develop well, although PRRC2A deficiency causes the abnormal upregulation of spermatogonia-specific genes. During the late prophase, PRRC2A is

highly expressed (Fig. 1b, c, S1) and exerts critical function, so multi-aspect defects occur in PRRC2A-null spermatocytes.

PRRC2A also enhances the translation of genes involved in meiotic cell division to facilitate the progression of meiotic metaphase. We found that PRRC2A interacts with multiple cofactors which are reported to degrade mRNA and/or promote translation initiation, including YBX1, YBX2, FXR1, PABPC1, and EIF4G3. Noted that mouse PRRC2A is a large protein with 2157 aa (isoform 2, NCBI ID: NP_001185973.1) or 2158 aa (isoform 1, NCBI ID: NP_064411.2). However, besides the GRE domain (domain enriched with glycine, arginine, and glutamic acid; <648 aa) used to bind m6A modification[21], the function of its BAT-N domain and other large regions is basically unknown. Additionally, PRRC2A was localized in RNA granules (chromatoid body) (Fig. 1c), this indicated the possible phase separation functions of PRRC2A. Thus, we speculate that PRRC2A potentially acts as a scaffold to recruit different transcript targets and corresponding cofactors in different developmental stages and locations, so that performs diverse regulations to its targets. Additionally, considering that many PRRC2A-binding transcripts do not contain m6A modification (2127 of 4081 PRRC2A-binding targets) (Fig. 5g), the large unknown regions of PRRC2A may also exert functions independent of m6A recognition to regulate mRNA targets directly.

Interestingly, many cytosolic m6A readers were reported to localize in RNA granules. In several human cell lines, YTHDF1, YTHDF2, and YTHDF3 are localized in stress granules under stress conditions, and YTHDF2 also exists with processing bodies[6,68,72]. IGF2BPs protect mRNAs from degradation in the processing body or store mRNAs in stress granules[17]. In spermatocytes, YTHDC2 was revealed to exist in RNA germ granules[7]. Further, PRRC2A is also co-localized with YTHDF2 in the RNA granule in HT-22 cells[21]. Here, we showed that PRRC2A is localized in the chromatoid body, a type of RNA granule specific for male germ cells. Chromatoid bodies contained several components of the NMD pathway[66] and are thought to mediate mRNA decay[67]. We speculate that PRRC2A may recruit target mRNAs to chromatoid bodies and promote the degradation. Additionally, PABPC1, FXR1, and EIF4G3 are also components of the stress granule and are revealed to interact with PRRC2A, results suggesting PRRC2A may partly interact with stress granules. This indicates that PRRC2A probably regulates target transcripts with other types of RNA granules under different conditions.

Here, our present study demonstrates that PRRC2A is required for the completion of male meiosis I by improving the degradation or translational efficiency of its mRNA targets. This research provides an important reference and theoretical basis for follow-up research and clinical practice in other processes in addition to spermatogenesis. Notably, although PRRC2A is expressed at low levels in other organs of adult mice, such as spleen and lung, and brain, PRRC2A may still play a role in special developmental stages or specific cells in these tissues, such as Pdgfrα- or NG2-positive cells in the embryonic brain[21]. Moreover, epidemiological studies reported that PRRC2A is associated with cancer[73,74], neurological diseases[75], autoimmune diseases[76], diabetes[77], and obesity[78]. Altogether, PRRC2A is widely involved in the regulation of various diseases and may be a potential clinical treatment target. More studies are needed to reveal the role of PRRC2A in other physiological and pathological processes.

## Methods

### Mice

All animal experiments were approved by the Chinese Ministry of Health national guidelines and performed following institutional regulations of Institutional Animal Care and Use Committee at the National Institute of Biological Sciences, Beijing. All mice in this study were C57BL6 strains. Mice were maintained under specific pathogen-free conditions of a 12 h light/dark cycle at controlled temperature

(20-25 °C) and humidity (50-70%) and were provided with food and water ad libitum in the Animal Care Facility at National Institute of Biological Sciences, Beijing.

Transgenic mice were generated using CRISPR/Cas9 technology. The sgRNAs were prepared using MEGAshortscript T7 Transcription kit (Ambion) according to the manufacturer's instructions. For *Prrc2a*$^{f/f}$ mice, DNA fragments containing exons 2–5 of the *Prrc2a* gene flanked by two loxP sites and two homology arms were used as donor templates. For *Prrc2a*-flag mice, DNA fragments containing the last coding exon (exon30) of the *Prrc2a* gene followed by a 3×flag tag and two homology arms were used as donor templates. After the co-incubation of Cas9 protein (NEB) and sgRNA, the Cas9-sgRNA complex and donor templates were injected into C57BL/6 zygotes. Injected zygotes were transferred into pseudo-pregnant CD1 female mice. The resulting founder mice were genotyped and mated with C57BL6 mice. For germ cell-specific knockout of *Prrc2a*, *Stra8*-Cre mice (Jackson Laboratory) were used. Briefly, *Prrc2a*$^{f/f}$ male mice were crossed with *Stra8*-Cre; *Prrc2a*$^{f/f}$ or *Stra8*-Cre; *Prrc2a*$^{f/\Delta}$ female mice, and descendants with *Stra8*-Cre were *Prrc2a*-cko mice. Only male mice were used in the current study. Sequence of gRNA and primers used for genotyping were listed in Supplementary Table 1.

### Histological analysis
Testes and epididymis were dissected and fixed in Davidson's Fluid (Formaldehyde: Ethanol: Glacial acetic acid: H2O = 6:3:1:10) overnight at 4 °C. Samples were dehydrated through an ethanol series (70%, 80%, 90%, 100% ethanol in H2O) and embedded in paraffin. The 5 μm sections were cut using a microtome (Leica RM2245) and mounted on adhesion microscope slides (CITOTEST). After deparaffinization and hydration, sections were stained with hematoxylin and eosin (H&E) following standard protocols. Images were acquired with the Olympus VS120 microscope.

### Chromosome spreads
Adult testes were dissected, and tunica albuginea was removed. 1.5–2 cm seminiferous tubules were incubated in 400 μL hypotonic extraction buffer (30 mM Tris-HCl pH 8.5, 50 mM sucrose, 17 mM citric acid, 5 mM EDTA) for 60 min at room temperature and minced in 100 mM sucrose/H$_2$O. 20 μL cell suspensions were dropped onto adhesion microscope slides (CITOTEST) spread with 20 μL fixation buffer (1% PFA and 0.15% Triton X-100) from a high place, fixed for 3 h at room temperature, and air-dried for 1 h in room temperature. Slides were washed three times in PBS (137 mM NaCl, 2.7 mM KCl, 10 mM Na$_2$HPO$_4$, and 1.8 mM KH$_2$PO$_4$) before immunohistochemical experiments.

### Immunostaining and TUNEL staining
Testes were dissected and fixed in 4% paraformaldehyde overnight at 4 °C. For paraffin sections, samples were treated as in histological analysis, and 5 μm sections were used for immunostaining. For frozen sections, samples were dehydrated in 30% sucrose/PBS and embedded in O.C.T compound (Tissue-Tek). 10 μm slices were cut using a cryostat (Leica CM1950) and dried overnight at 42 °C. After washing three times with PBS, paraffin or frozen sections were subjected to antigen retrieval with sodium citrate buffer (10 mM sodium citrate, 0.05% Tween-20, pH 6.0) or Tris-EDTA buffer (10 mM Tris base, 1 mM EDTA, 0.05% Tween-20, pH 9.0). Sections were blocked in ADB (1% normal donkey serum, 0.3% BSA, 0.05% Triton X-100) for 1 h at room temperature and incubated with primary antibodies in ADB overnight at 4 °C. After washing three times with PBST (PBS with 0.1% Tween-20), slides were incubated with secondary antibodies in ADB for 1 h at room temperature and washed another three times followed by staining with 1 μg/mL 4',6-Diamidino-2-phenylindole (DAPI) (Invitrogen, D3571). For staining of chromosome spreads, slides were blocked with ADB and incubated with primary antibodies, secondary antibodies, and DAPI

(Invitrogen, D3571) as above. For detecting the acrosome, 1 μg/mL PNA (Sigma, L7381) was used to stain sections for 20 min at room temperature. For TUNEL staining, assays were performed using In Situ Cell Death Detection Kit (Roche, 11684795910) following the manufacturer's instructions. Images were acquired using the confocal microscope Zeiss LSM800 (Zen 2.3 (blue edition) software) or Nikon A1-R (NIS-Elements AR 5.21.00 software) and microscope Olympus VS120 (OLYMUPUS VS-ASW 2.9 software).

### Western blot
Testes were lysed in RIPA buffer (Sigma, R0278) with 1: 10 protease inhibitor cocktail (Roche, 04693116001). The lysate was centrifuged at 13,000 rpm 4 °C for 20 min. The protein concentration of supernatants was measured with Quick Start$^{TM}$ Bradford 1 x Dye Reagent (BIO-RAD, 500-0205). For isolated germ cells, they were lysed with 1x loading buffer (2% SDS, 10% Glycerol, 50 mM Tris-HCL, 1% β-mercaptoethanol, and 0.05% bromophenol blue dye, PH6.8) directly. The equal quality of proteins of each sample was separated by sodium dodecyl sulfate-polyacrylamide gel electrophoresis (SDS-PAGE) and transferred to polyvinylidene difluoride (PVDF) membranes (Millipore, IPVH00010). Membranes were blocked with 5% skim milk in TBST (20 mM Tris, 150 mM NaCl, pH 7.6 with 0.05% Tween-20) and incubated with the primary antibody in 5% skim milk/TBST. After three times of wash with TBST, membranes were incubated with HRP-conjugated secondary antibody in 5% skim milk/TBST for 1 h at room temperature followed by another three times of washing with TBST. ECL reagents (BIO-RAD, 170-5060 or NCM Biotech, P10300B) were added onto membranes, and signals were detected by XBT X-ray film (Carestream, 6535876). All uncropped and unprocessed scans of western blots were supplied in Supplementary Fig. 8 in the Supplementary Information.

### Antibody
Antibodies used in the immunostaining and WB were listed in Supplementary Table 2.

### RNA purification and quantitative real-time PCR
Tissues or cells were homogenized in TRIZOL (Invitrogen, 15596026) and chloroform was added to extract twice. After centrifugation, the upper aqueous phase was transferred into isopropanol followed by another centrifugation to collect RNA pellets. Glycogen (Thermo Scientific, R0551) was used when precipitating a small amount of RNA. Pellet was washed two or three times using 75% ethanol, air-dried, and solubilized in nuclease-free water. Reverse transcription was performed using PrimeScript® RT reagent kit with gDNA Eraser (TaKaRa, RR047A) according to the manufacturer's instructions. Quantitative real-time PCR (qPCR) was performed using SYBR Green master mix (TaKaRa, DRR420A) and Bio-Rad CFX96 Real-Time System (Bio-Rad CFX Manager 3.1 software). Relative mRNA expression was measured using the Delta-Delta CT method, and *Rpl6* was used for normalization. Primers used were listed in Supplementary Table 1.

### Immunoprecipitation and Mass spectrometry
For immunoprecipitation, four P20 testis was homogenized in 1 mL ice-cold lysis buffer (150 mM NaCl, 10 mM HEPES pH 7.6, 2 mM EDTA, 0.5% NP-40, 0.5 mM 1,4-dithiothreitol (DTT), 1:10 protease inhibitors cocktail (Roche, 04693116001), 100 U/mL Recombinant RNasin® Ribonuclease Inhibitor (Promega, N2515)) and incubated on ice for 20 min to lyse the tissue. For RNase treatment, RNasin and DTT were replaced by 10 mg/mL RNase A (Tiangen). The lysate was centrifuged at 13,000 ×g for 20 min at 4 °C and supernatants were collected. 100 μL supernatants were saved as input and mixed with 25 μL 5 x loading buffer (10% SDS, 50% Glycerol, 250 mM Tris-HCL, 5% β-mercaptoethanol and 0.1% bromophenol blue dye, pH 6.8). 30 μL Protein G magnetic beads (Invitrogen, 10004D) conjugated with 1.5 μg mouse antibody to Flag (Sigma, F1804) or mouse IgG isotype control

(CST, 5415 S) were added into 300 μL supernatants and incubated overnight at 4 °C. Supernatants were discarded and beads were washed 8 times with 1 mL ice-cold NT2 buffer (200 mM NaCl, 50 mM HEPES pH 7.6, 2 mM EDTA, 0.05% NP-40) at 4 °C. For RNasin treatment, 0.5 mM DTT and 40 U/mL RNase inhibitor were added. For the RNase A treatment group, after 3 times washes, beads were treated with NT2 buffer supplemented with 10 mg/mL RNase A for 20 min at room temperature and washed with NT2 buffer another 5 times. Then, beads were boiled in 1x loading buffer (2% SDS, 10% Glycerol, 50 mM Tris-HCL, 1% β-mercaptoethanol, and 0.05% bromophenol blue dye, pH6.8) for 10 min at 95 °C. After centrifugation, supernatants were subjected to Western blot detection.

For the identification of interaction proteins by MS, IP was performed as above with some modifications. In brief, eight P20 *Prrc2a*-flag and wild-type testes were used and supernatants of lysate were incubated with 60 μL Protein G magnetic beads (Invitrogen, 10004D) conjugated 3 μg mouse antibody to Flag. After 8 times washes, beads were eluted using 0.2 mg/mL Flag peptide (Sigma, F4799) in NT2 buffer for 1 h at 4 °C. Samples were separated on SDS-PAGE followed by silver staining (Sigma, PROTSIL1). The stained proteins were destained and in-gel digested with trypsin (10 ng mL-1 trypsin, 50 mM ammonium bicarbonate, pH 8.0) overnight at 37 °C. Peptides were extracted with 5% formic acid/50% acetonitrile and 0.1% formic acid/75% acetonitrile sequentially. The extracted peptides were separated by an analytical capillary column (50 μm × 10 cm) and sprayed into an LTQ ORBITRAP Velos mass spectrometer (Thermo Fisher Scientific, San Jose, CA, USA) equipped with a nano-ESI ion source. Identified peptides were searched in the IPI (International Protein Index) Mouse protein database on the Mascot server (Matrix Science Ltd, UK).

### Isolation of late meiotic spermatocytes

The procedure was modified from the previous study[55]. Testes were dissected and tunica albuginea was removed in PBS. Seminiferous tubules of one control testis or two *Prrc2a*-cko testes were dispersed with tweezers and incubated in 10 mL DMEM medium containing 1 mg/mL Collagenase IV (YEASEN, C3125030) and 0.1 mg/mL DNase I (YEASEN, D2122070) for 25 min at 34 °C with rotation. Tubule fragments were collected by gravity settlement and washed twice with 10 mL PBS. Then, tubules were digested in 10 mL 0.05% Trypsin/EDTA (Gibco, 25300062) containing 0.1 mg/mL DNase I for 8 min at 34 °C and were gently pipetted up and down to disperse germ cells. 1 mL FBS was added and the cell suspension was passed through nylon mesh with 40 μm pore size (FALCON, 352340). After centrifuging of 300 × g for 5 min, germ cells were washed by 1 mL DMEM containing 10% FBS and resuspended by 4 mL DMEM containing 10% FBS. Then, 4 μL 10 mg/mL Hoechst 33342 (Sigma, B2261) was added, and cells were stained for 60 min at 34 °C with rotation. Before sorting, 8 μL 1 mg/mL propidium iodide (Invitrogen, P3566) was added, and the cell suspension was filtered by 40-μm nylon mesh another time. Cell suspensions were sorted by BD FACSAria Fusion-II with the 70 μm nozzle using BD FACSDiva software (version 8.0.3). 355 nm laser was used to excite Hoechst 33342 and fluorescence was recorded with a 450/40 nm band-pass filter (Hoechst blue) and a 635 nm long filter (Hoechst red). The gating strategy referred to the previous study[55]. Spermatocytes were collected in DMEM containing 10% FBS for subsequent experiments. Flowjo software (vX.0.7) was used for data analysis.

### RNA-seq

Purified mRNAs from isolated spermatocytes of adult control and *Prrc2a*-cko testes and were used to construct libraries using NEBNext Ultra II DNA Library Prep Kit for Illumina (NEB, E7645L) in the sequencing center at National Institute of Biological Sciences, Beijing. The libraries were sequenced on the Illumina HiSeq 2500 platform using the single-end 75 bp sequencing strategy.

Raw sequencing reads were trimmed to remove low-quality bases by Trim Galore (version 0.6.4)[79] in single-end mode. Then the trimmed reads were aligned to the mouse reference genome assembly build GRCm38_68 (mm10) using STAR (version 2.7.3a)[80] with default parameters. The mapped reads were annotated to ensemble gene exons (Mus_musculus.GRCm38.99.gtf) and counted for each gene by featureCounts (version 2.0.0)[81], the expression level of each gene was quantified as reads per kilobase of transcript per million reads mapped (RPKM). Differentially expressed genes (DEGs) between *Prcc2a* knockout and control were identified based on read counts by using DEseq2 (version 1.30.1)[82] with the following cutoffs: fold change > 1.5, adjusted *p*-value < 0.05, mean RPKM > 1, each RPKM > 0. Heatmap plots were plotted on http://www.bioinformatics.com.cn, a free online platform for data analysis and visualization.

### Ribo-seq

Cells were treated with 100 μg/mL cycloheximide (CST, 2112) to block translational elongation during the whole process of spermatocyte isolation using FACS. Isolated spermatocytes were then frozen with liquid nitrogen and stored at −80 °C for subsequent experiments. For ribosome footprints (RFs) recovery, samples were dissolved in 400 μL of lysis buffer and the ribosomal profiling technique was carried out as reported previously[83], with a few modifications as described below. The extracts were incubated on ice for 10 min and were triturated ten times through a 26-G needle. The lysate was centrifuged at 20,000 × g for 10 min at 4 °C, and the supernatant was collected. Then, 10 μL of RNase I (NEB) and 6 μL of DNase I (NEB) were added to 400 μL of lysate, which was then incubated for 45 min at room temperature. Nuclease digestion was stopped by adding 10 μL of SUPERase·In RNase inhibitor (Ambion). Next, 100 μL of digested RFs were added to the equilibrated size exclusion columns (illustra MicroSpin S-400 HR Columns; GE Healthcare; catalog no. 27-5140-01) and centrifuged at 600 g for 2 min. Then, 10 μL 10% (wt/vol) SDS was added to the elution, and RFs with a size >17 nt were isolated according to the RNA Clean and Concentrator-25 kit (Zymo Research; R1017). rRNA was removed using the method reported previously[84]. Briefly, short (50–80 bases) antisense DNA probes complementary to rRNA sequences were added to the solution containing RFs, then RNase H (NEB) and DNase I (NEB) were added to digest rRNA and residual DNA probes. Finally, RFs were further purified using magnet beads (Vazyme). Ribo-seq libraries were constructed using NEBNext® Multiple Small RNA Library Prep Set for Illumina® (catalog no. E7300S, E7300L) following the manufacturer's instructions and were sequenced using Illumina HiSeqTM X10.

We followed the pre-processing procedure of the ribosome profiling data as described previously[85]. Briefly, the Trim Galore (version 0.6.4)[79] was used to trim the 3' adapter in the raw reads. Low-quality reads with Phred quality score >20 were removed. Next, the trimmed reads were aligned to abundant sequences (including rRNA, tRNA, and mtRNA) by using Bowtie2 (version 2.3.5.1)[86] with no mismatch allowed, and mapped reads were discarded. Then the unmapped reads were aligned to the mouse reference genome assembly build GRCm38_68 (mm10) using STAR (version 2.7.3a)[80] as did in mRNA processing, and two mismatches were allowed in this step. To reduce the technical noise of ribosome profiling, in the quantification of ribosome-protected mRNA fragments (RPFs), the following filters were processed. Firstly, the Ribo-seq reads with length between 26 and 34 nt were selected. Besides, the multiple aligned reads were discarded, and only the uniquely mapped reads to the coding regions were retained. Third, reads aligned to the first 15 and last 5 codons were excluded. After filtrations, the quantification of RPFs was processed by featureCounts (version 2.0.0)[81]. The counts matrix of RPFs were combined with expression matrix, and implemented into Xtail package (version 1.1.5)[85] to calculate the differential translation efficiencies. The threshold for differential translational efficiency was fold change > 1.5, *p*-value < 0.05.

Heatmap plots were plotted on http://www.bioinformatics.com.cn, a free online platform for data analysis and visualization.

## Functional enrichment analysis

DTGs and DTEGs were performed for functional enrichment analysis by using the online tool Metascape (http://metascape.org)[87]. Gene Ontology biological processes (BP) pathways were selected as ontology sources. Terms with $p$-value $< 0.01$ were retained as significant enrichment. Gene set enrichment analysis for the cell-type-specific gene sets and gene sets annotated with GO terms "meiotic cell cycle", "meiotic nuclear division", and "spindle" was performed using GSEA software (version 4.1.0)[88] with 1000 gene set permutations

## PRRC2A RIP-seq

The procedure was modified from the previous study[21]. Thirty-two testes from P20 *Prrc2a*-flag mice were dissected and homogenized in 8 mL ice-cold lysis buffer (150 mM NaCl, 10 mM HEPES pH 7.6, 2 mM EDTA, 0.5% NP-40, 0.5 mM 1,4-dithiothreitol (DTT), 1:10 protease inhibitors cocktail (Roche, 04693116001), 100 U/mL Recombinant RNasin® Ribonuclease Inhibitor (Promega, N2515)) and incubated on ice for 20 min to lyse the tissue. The lysate was centrifuged at 13,000 × $g$ for 20 min at 4 °C supernatants were collected, then repeat centrifugation twice. Protein concentration was measured using Quick Start™ Bradford 1x Dye Reagent (BIO-RAD, 500-0205) and the sample was diluted to make the concentration not >5 mg/mL. 200 μL sample was used as input and RNA was purified using the TRIZOL method. The remaining sample was incubated with 6 μg mouse IgG isotype control (CST, 5415 S) for 1 h at 4 °C and 75 μL protein G magnetic beads (Invitrogen, 10004D) for 1 h at 4 °C to preclear. The beads were removed and 250 μL protein G magnetic beads conjugated with 20 μg mouse anti-Flag antibody (Sigma, F1804) were added into supernatants and incubated overnight at 4 °C with rotation. Supernatants were discarded and beads were washed 8 times with 1 mL ice-cold NT2 buffer (200 mM NaCl, 50 mM HEPES pH 7.6, 2 mM EDTA, 0.05% NP-40). Next, beads were washed twice with 1 mL ice-cold DNase buffer (10 mM Tris-HCL, 2.5 mM MgCl$_2$, 0.5 mM CaCl$_2$, 0.05% NP-40, pH 7.6) at 4 °C and incubated in 500 μL DNase buffer containing 100 U/mL DNase I (NEB, M0303S) for 10 min at 37 °C. After that, beads were washed twice with 1 mL ice-cold MNase buffer (50 mM Tris-HCL, 5 mM CaCl$_2$, 100 μg/mL BSA, pH 7.9) and digested in MNase buffer containing 1 U/mL MNase (NEB, M0247S) for 10 min at 37 °C Then, supernatants were discarded and beads were washed twice with 1 mL 1×PNK + EGTA buffer (50 mM Tris-HCl pH 7.5, 20 mM EGTA, 0.5% NP-40, pH 8.0) immediately at 4 °C and twice with 1 mL NT2 buffer 4 °C. Collected beads were digested with 4 mg/mL proteinase K (Roche, 03115828001) in 200 μL 1×PK buffer for 40 min at 55 °C. RNA was extracted from supernatants using the TRIZOL method. Input and RIP samples were subjected to rRNA removal using rRNA Depletion Kit (NEB, E6310S) according to the manufacturer's instructions. The resulting RNA samples were examined by Agilent 2100 Bioanalyzer and used to generate the library using NEBNext® Ultra™ II DNA Library Prep Kit for Illumina® (NEB, E7645L). Sequencing was performed on HiSeq X Ten System with 150 bp paired-end-sequencing reactions. Biological replicates were performed and samples for each biological replicate were collected from different animals.

## PRRC2A RIP-seq and m6A MeRIP-seq analysis

The raw sequencing reads were trimmed to remove the adapter sequences and low-quality bases by Trim Galore (version 0.6.4)[79]. Reads that were >35 bps were retained and aligned to the mouse genome (mm10) by using Bowtie2 (version 2.3.5.1)[86]. Then uniquely aligned reads with mapping quality score $\geq 20$ were kept for downstream peak-calling. Peak-calling was processed by MACS2 (version 2.1.2)[89] for each biological replicate by comparing with input sample

with parameters: "–keep-dup all –nomodel -q 0.05". The peaks derived from two replicates were merged and the overlapped peaks were retained as high-confidence binding regions of PRRC2A or m6A modification regions. The m6A MeRIP-seq data was referred to the published dataset (GEO: GSE102346)[11]. The overlapped peaks between PRRC2A RIP-seq and MeRIP-seq were obtained using BED-Tools' intersect (version 2.28.0)[90]. The coverage profiles of RIP-seq were counted and visualized by igvtools (version 2.9.4)[91]. Peak with counts >1 and counts ratio (IP versus Input) >1 were used for the next analysis.

## Motif Identification

We selected all peaks ranked by $q$-value to investigate the motifs enriched in these regions by using HOMER (version 4.11.1)[92]. To obtain high-quality motifs, repeat sequences were masked, motif length was restricted to 5–10, and the finding region was limited to 100 nt relative from the peak center. Furthermore, background sequences for each peak were generated on mRNA sequences by BEDTools' shuffleBed (version 2.28.0)[90] command to remove the background signal of selected genes.

## PRRC2A RIP-qPCR

RNA immunoprecipitation of PRRC2A was performed as procedures in PRRC2A RIP-seq with some modifications. Four P20 testes from P20 wild-type mice were used and homogenized. Tissue lysate was pre-cleared with 2 μg mouse IgG isotype control (CST, 5415 S) and 25 μL protein G magnetic beads (Invitrogen, 10004D). The precleared lysate was incubated with 50 μL protein G magnetic beads conjugated mouse IgG isotype control (IgG group) or mouse antibody of Flag (Sigma, F1804)(Flag group). Then, beads were washed with NT2 buffer, treated with DNase I (NEB, M0303S) and MNase (NEB, M0247S), and digested with proteinase K (Roche, 03115828001). Supernatants were collected and RNA was extracted using the TRIZOL method. RNA samples were reverse transcribed using PrimeScript® RT reagent kit with gDNA Eraser (TaKaRa, RR047A) and detected by qPCR using SYBR Green master mix (TaKaRa, DRR420A). Three biological replicates were performed and samples for each biological replicate were collected from different animals. The recovery rate of specific transcripts was calculated by comparing the relative mRNA abundance of target genes in the IP group with the input group, and the level of *Gapdh* was used to normalize IP and IgG groups. Primers used were listed in Supplementary Table 1.

## MeRIP-qPCR

RNA samples were extracted from four P20 wild-type testes and purified using the MiRNeasy mini kit (Qiagen, 217004) with DNase set (Qiagen, 79254). Purified RNAs (50 μg) were broken into ~300 nt fragments using RNA Fragmentation Reagents (Invitrogen, AM8740) by 30 s incubation at 94 °C. Fragmented RNAs were collected and purified by ethanol precipitation. Next, the RNA sample was incubated with 40 μL protein A magnetic beads (Invitrogen, 10002D) conjugated with 2 μg Rabbit IgG isotype control (CST, 2729 S) or rabbit antibody to m6A (Synaptic Systems, 202003) in IPP buffer (150 mM NaCl, 0.1% NP-40, and 10 mM Tris-HCl, pH 7.4, 1 mM 1,4-dithiothreitol (DTT), 40 U/mL Recombinant RNasin® Ribonuclease Inhibitor (Promega, N2515)) for 4 h at 4 °C. Then, beads were washed three times with IPP buffer and digested by Proteinase K (Roche, 03115828001). Bound RNA was extracted from supernatants using the TRIZOL method and reverse transcribed using PrimeScript® RT reagent kit with gDNA Eraser (TaKaRa, RR047A). Relative mRNA abundance of target genes was detected by qPCR using SYBR Green master mix (TaKaRa, DRR420A). Three biological replicates were performed and samples for each biological replicate were collected from different animals. The recovery rate of specific transcripts was calculated by comparing the relative mRNA abundance of target genes in the IP group with the input

group, and the level of *Gapdh* was used to normalize IP and IgG groups. Primers used were listed in Supplementary Table 1.

## Phenotype characterizing and quantification analysis

For the staging of seminiferous tubules, due to the lack of late-stage spermatids in *Prrc2a*-cko testes, stages of seminiferous tubules were determined based on the expression pattern of γH2AX and the morphology and composition of germ cells (Supplementary Fig. 3d). In adult *Prrc2a*-cko testes, tubules with early pachytene spermatocytes (cells close to the base of the lumen and with nuclear dense punctate γH2AX signal) and round spermatids were determined as stage I–III; tubules with mid-pachytene spermatocytes (cells relatively far from the base of the lumen and with nuclear dense punctate γH2AX signal) and round spermatids (little or no) were determined as stage IV–VII; tubules with leptotene spermatocytes (cells located at the base of the lumen and with nuclear γH2AX signal), late-pachytene spermatocytes (cells far from the base of the lumen and had large nuclei with dense punctate γH2AX signal) and no spermatid were determined as stage VIII–X; tubules with zygotene spermatocytes (cells located at the base of the lumen and with nuclear dense but not completely concentrated γH2AX signal) and diplotene spermatocytes (cells far from the base of the lumen and had large nuclei with dense punctate γH2AX signal) were determined as stage XI–XII. Only round and characteristic tubules were considered for followed analysis. Germ cells far from the base of the lumen and with condensed chromosomes aligning or aligned on the equatorial plate were determined as metaphase spermatocytes.

For phenotype characterization using histology and immunofluorescence experiments, three biological replicates were used and representative images were shown. For quantification of staining of TUNEL and pH3, at least 3 testes from different mice were used. For quantification of the immunofluorescence intensity of POL II in the XY region, intensity in the XY region and the whole cell was measured and exported as 8-bit intensity values. The intensity of the area except for the XY body was calculated and used to normalize the intensity in the XY region. For quantification of MLH1 and DMC1 foci in chromosome spreads of prophase spermatocytes, foci along chromosomes were counted. For quantification of the immunofluorescence intensity of γ-tubulin and CEP192, the intensity at the spindle pole was measured and exported as 8-bit intensity values. The width and length of the alignment and spindle and the intersection angle between the spindle polarity axis and equatorial plate were measured in P60 testes sections stained with α-tubulin and γ-tubulin. Fiji software[93] was used to measure the fluorescence intensity, in images acquired with the confocal microscope. Scatter plots and bar plots were created and analyzed with GraphPad Prism 6 (GraphPad Software). All data are presented as the means ± SEM. Significance difference was tested with Two-sided student' s *t*-test (ns $p > 0.05$; *$p < 0.05$; **$p < 0.01$; ***$p < 0.001$).

## Statistics and reproducibility

For all histology, immunofluorescence, western blot, immunoprecipitation, and qPCR experiments, we performed at least three independent biological replicates. All uncropped and unprocessed scans of the western blots were supplied in Supplementary Fig. 8 in the Supplementary Information.

## Reporting summary

Further information on research design is available in the Nature Portfolio Reporting Summary linked to this article.

## Data availability

All sequencing data generated in this study have been deposited in the Genome Sequence Archive (GSA) (https://ngdc.cncb.ac.cn/gsa/) of China National Center for Bioinformation-National Genomics Data Center (CNCB-NGDC) under accession code: CRA005170. Source data are provided with this paper.

## Code availability

The custom scripts used in this study will be available on request from the corresponding authors.

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

## Acknowledgements

We thank Taojun Zhang for his help with mouse breeding, genotyping, histology analysis, and immunohistochemistry experiment. We thank Rong Wu for suggestions for this study and instructions for the PRRC2A RIP-seq experiment. We thank Hui Han and Xiaofeng Feng for the mass spectrometry experiment. We thank John Hugh Snyder for the professional editorial advice and polishing of the manuscript. This work was supported by the National Key R&D Program of China [2018YFC1003102 to C.Z.] and the Youth Innovation Promotion Association of Chinese Academy of Sciences [2020104 to C.Z.].

## Author contributions

F.W. conceived and supervised the project. F.W. and X.T. designed the experiments. Y.Z. and P.J. generated transgenic mice. X.T. performed mouse breeding, experiments, and related quantification. C.Z. processed the high-throughput RNA sequencing data. C.Z. and X.T. analyze the sequencing data. X.T., C.Z., and F.W. wrote the manuscript. X.T. and C.Z. contributed equally to this work.

## Competing interests

The authors declare no competing interests.
