## [Peer Review File · Nature Communications]

The m6A reader PRRC2A is essential for meiosis I completion during spermatogenesisREVIEWER COMMENTS

Reviewer #1 (Remarks to the Author):

In this manuscript by Tan et al., titled "The m6A reader PRRC2A is essential for successful completion of meiosis I in spermatogenesis by promoting mRNA stability", the authors describe an essential role for PRRC2A during spermatogenesis. They demonstrate that PRRC2A is critical for spindle organization and chromosome segregation during metaphase I of meiosis. They further show that PRRC2A binds centrosome-encoding transcripts among others, and directly/indirectly affects their steady-state levels. The findings reported here are novel and experiments conducted are extensive. Few comments are listed below.

Major comments:

1) The authors report that there is an MSCI defect in *Prcc2a* cKO based on increased signal intensity of Pol II at/near XY. While this is consistent with the upregulation of XY genes in whole testis RNA seq at P20 in *Prcc2a* cKO, other data do not fully support that the upregulation stems from an MSCI defect. MSCI defects lead to a mid-pachytene arrest (see Royo et al. 2010 *Curr Biol* or Turner 2015 *Annu Rev Genet*). The authors observe few TUNEL-positive pachytene cells consistent with this (Fig 1G), but this should be quantified and connected to the small (although not significant) increase in TUNEL staining seen at ~stages IV-VII (Fig 1H). And MSCI defects lead to loss of gamma-H2AX accumulation on XY. This does not appear to be the case in Fig 2A, or is this observed at a low frequency? To test whether there is an MSCI defect, the authors would need to examine localization of silencing/MSCI factors on XY (e.g., gamma-H2AX; there are several others that could be tested although this is not necessary: *HORMAD2*, *BRCA1*, *ATR*, *TOPBP1*, *MDC1*). And/or the authors could perform RNA FISH of an XY gene to show that inappropriate gene XY expression is occurring specifically in pachytene cells.

2) The authors state that PRRC2A binds and stabilizes centrosome-encoding transcripts, and show a stabilization function for select targets by performing half-life measurements in cultured meiotic cells. For a more genome-wide examination, they examine the overlap between down regulated transcripts in *Prcc2a* cKO and PRRC2A RIP targets plus m6A-modified transcripts and detect ~380 transcripts. Do the upregulated transcripts show a smaller overlap, consistent with the principal function being to promote stability? Similarly, the comparison made with centrosome-encoding transcripts excludes upregulated transcripts. How many centrosome-encoding transcripts overlap with the PRRC2A RIP plus m6A-modified plus upregulated transcripts? These comparisons should be explained in the text and contrasted with those already mentioned.

Minor comments:

1) In lines 174-175, the authors state that XY asynapsis triggers impaired MSCI (and they cite a *Mettl3* study). I am not sure what the authors mean by this, as the asynapsis of X and Y is connected to normal MSCI. Please could the authors check/clarify this statement and cite appropriate refs.

2) Can the authors be certain that the interactions observed between PRRC2A and YBX1/YBX2 are not indirect through RNA, since interactions tested via IP are lost after RNase treatment (Fig 6C). This should be addressed in the text and the PRRC2A interaction statements should be modified accordingly.

3) Wojtas et al. *Mol Cell* 2017 reference should be included when testis m6A studies are described (line 70).

4) Figure S1B is hard to see. I suggest providing magnified insets.

5) In Fig 1H the annotation is probably meant to be $p=0.054$ and gamma-H2AX is incorrectly written

throughout.

Reviewer #2 (Remarks to the Author):

The manuscript "The m6A reader PRRC2A is essential for successful completion of meiosis I in spermatogenesis by promoting mRNA stability" by Tan et al show that the m6A reader PRRC2A, using a male germ cell-specific *Prrc2a* knockout mouse model, is necessary for meiosis I progression and propose that PRRC2A promotes the stability of transcripts of centrosome-contained proteins through binding them. This work aims to enhance our understanding of an important regulatory process (posttranscriptional regulation during meiosis I).

Major concerns:

A lynchpin argument for this report is that "Volcano plot of RNA-seq data shows the transcriptome difference in P20 control and *Prrc2a*-cko testes. p-value = 0.01 and 1.2-fold expression difference are used as the cutoff" (Lines 1120-23) and that "we performed polysome analysis and found no significant difference in overall translation status between P20 *Prrc2a*-cko and control testes." (Lines 328-329)

- 1) The experiments for RNA-seq and polysome analysis in this manuscript used whole testes. Thus, the data presented are not solid. This reviewer would like to require the authors use purified spermatogenic cells for RNA-seq and subsequent polysome analysis.
- 2) The cutoff value for the expression difference is too low. This reviewer would like to require the authors to provide corresponding results in Figure 4 with a relative higher cutoff value (at least, for example 1.5).

Minor issues:

1. The manuscript needs careful editing to improve the use of English grammar.
2. The author indicated in Fig 2b that *Prrc2a*-cKO spermatocytes showed XY asynapsis in late-pachytene, but the MLH1 foci number in Fig S4F has no significant change. Could the authors present a probable explanation for these data?

Reviewer #3 (Remarks to the Author):

In this manuscript, Tan and colleagues explore the role of the m6A reader PRRC2 in spermatogenesis. Using a powerful in vivo cKO mice model, authors demonstrate the role of this protein in the regulation of the correct chromosome alignment during meiosis. Using a combination of RNA sequencing and RIP-seq, MeRIP-seq authors propose a model in which the absence of PRRC2 destabilize some of the centrosome-related mRNAs leading to chromosomal instability. This manuscript shed light on PRRC2 functions as an m6A reader in the spermatogenesis complex, linking the strong phenotype observed in these mice with the function of one m6A reader. The manuscript presents a strong dataset and provide and significant advance in the correct comprehension of the role of this protein in mRNA regulation. However, some of the affirmations need further analysis and or/controls to support the molecular mechanism they describe, and the link between m6A methylation and the phenotype observed should be demonstrated to a greater extent.

Major concerns

1. Deletion of PRRC2 seems to have a role in the stabilization of the transcripts analysis of the transcriptome reveals an important general transcript downregulation (Fig4). However differential expression analysis is not restrictive enough; a 1.2 fold change (log₂ fold 0,26) might only reflect stochastic differences among the different replicates with no biological consequences. I understand the

need of lowering the value of fold to detect small changes, otherwise screened because the analysis is made in complex tissues. But even in this case, variation among the replicates must be assessed and fold change adjusted accordingly to it in the differential expression analysis. Do the protein level of the analyzed mRNAs change in p20 testes in cKO mice?

2. Heatmaps showing X- and Y- linked genes seem to show a high variance in the heatmap with big differences among the replicates (Fig S6A), and in fact, some of the representative genes picked to confirm show no statistically significant increase or no biological significant increase. A wider panel of genes must be analyzed to confirm the general de-repression of X- and Y-linked genes.

3. In figure 6D, stricter differential expression analysis is needed together with statistical significance of the overlap. Also, overlapping controls with the upregulated genes will be useful to discern if this overlap is significant or not. That concern can also be applied to Sup. figure 6E.

4. Authors suggest that PRRC2 promote the stability of mRNAs recognizing m6A but only one-third of PRRC2 RIP-seq peaks overlaps m6A peaks, and only 20% of those overlapping peaks are downregulated (in a low restrictive analysis). Some of the genes they analyze do not get downregulated despite having been immunoprecipitated in m6A and PRRC2 RIP (Cep152 and Cep192 Figures 6F and 6K & L made in p20 testis). Authors should justify these low occurrence numbers in the discussion and other possible effects that might be affecting the regulation of these genes, and if possible, demonstrate the link between m6A methylation and the downregulation of these genes.

5. PRCC2A RIP and M6A meRIP data (Figure 5K and L must be represented as % of input with proper controls (Dazl, GAPDH or m6A positive controls) in the same scale. Referencing data to IgG is not valid as it is a random low number and any reference to them will produce random results.

6. Moreover, authors should consider performing CLIP assays to distinguish real interactions occurring in the cell from those just produced by biochemical affinity or by secondary interactions.

7. PRRC2A binds YBX1 and YBX2 in an RNA dependent fashion, suggesting they are bound to the same RNAs but not form a complex, however, PRRC2A binds PABPC1 in an RNA independent manner indicating they form part of a complex. Is it the stability effect the authors observe depending on the methylation of the RNA or is it due to the alteration of the interaction of PABPC1 with the RNA which will produce de-stabilization of the mRNA?

Minor concerns

1. Some of the figures do not follow the text order making it a bit difficult to follow the logic. There are also some figure citations wrong placed in the text such as "figure 3I" in line 298.

2. Is endogenous PRRC2A-FLAG less expressed than the wt? It seems despite the FLAG being more powerful antibody images are much more exposed. Ratios between Input IP with FLAG and PRRC2A antibodies are quite different.

Dear reviewers,

Thank you for your careful review and constructive suggestions regarding our manuscript entitled “The m6A reader PRRC2A is essential for successful completion of meiosis I in spermatogenesis by promoting mRNA stability” (NCOMMS-21-41916). On behalf of my co-authors, we appreciate you very much for your positive comments.

Those comments are all valuable and very helpful for revising and improving our paper, as well as the important guiding significance to our research. We have studied the comments carefully and have heavily revised our experiments and manuscript which we hope to meet with approval. In the manuscript, revised portions are marked with yellow color for modified or added sentences, or are marked with green color and strikethrough for deleted sentences. We also did lots of language editing which was presented in red text in the **“revised manuscript showing polishing”**, hoping to improve the English grammar and readability. We also provided a **“revised manuscript hiding polishing”** for a better reading experience. In the point-to-point response document, our responses were presented in blue text and we used **line numbers in the manuscript hiding polishing**.

Notably, we replaced **RNA-seq and polysome profiling data using P20 testes** (lines 1413-1420, lines 1422-1444 (picture D-F), lines 1536-1546 (picture A, B, E, F), and lines 1548-1558) with **RNA-seq and Ribo-seq data using sorted spermatocytes** (Fig. 5, 6, 7A, S6A, S6B, S6E) according to your and reviewers’ suggestions (lines 303-304), and the conclusion of our manuscript has greatly changed based on these new sequencing data. And we also changed the title to “The m6A reader PRRC2A is essential for successful completion of meiosis I in spermatogenesis”.

Detailly, for the sequencing of spermatocytes, we initially planned to purify spermatocytes from P20 control and *Prrc2a*-cko testes using FACS, but surprisingly, we found that late prophase spermatocytes were dramatically less in *Prrc2a*-cko P20 testes than in controls (Fig. S4C, lines 209-214), phenotypes we did not notice before. So, we decided to perform RNA-seq and Ribo-seq using spermatocytes from adult testes (lines 303-304). The sequencing data revealed a quite different pattern of the transcriptome between using P20 testes and using sorted spermatocytes (lines 297-501). Thus, these data promoted us to dismiss the RNA-seq and polysome analysis data for P20 testes, and study functions of PRRC2A only based on the sequencing data of sorted spermatocytes.

Sequencing data showed that PRRC2A downregulates the RNA abundance or upregulates the translation efficiency of its mRNA targets (Fig. 5, lines 297-385). Specifically, PRRC2A-null spermatocytes fail to turn off the expression of spermatogonia-specific genes and cannot maintain the normal expression pattern of spermatocytes (Fig. 6, lines 386-425). For several genes involved in meiotic cell division, PRRC2A recognizes their transcripts and improves the translation efficiency

(Fig. 7A-G, S6F, lines 426-501). Further, we found that PRRC2A interacts with several cofactors regulating mRNA metabolism and translation (YBX1, YBX2, PABPC1, FXR1, and EIF4G3) (Fig. 7H, S7B, lines 507-543). Thus, we believed that PRRC2A is not an mRNA stabilizer but potentially promotes the decay or translation of its mRNA targets

Additionally, we found that PRRC2A deficiency also caused delayed metaphase entry (Fig. 3B-D, lines 232-243) with decreased protein abundance of CDK1 (core protein of MPF) (Fig. 4C, lines 290-294) and led to downregulated protein abundance of EIF4G3 (Fig. 7J, lines 516-521). Considering recent studies showed that YTHDC2 regulates spermatogenesis independent of m6A recognition, we added the corresponded statement in introduction (lines 85-87) and discussion (lines 566-567).

And unexpectedly, we found the RNA abundance of *Gapdh* were noticeable upregulated in PRRC2A-null spermatocytes (shown in below table), so we switched the normalization gene from *Gapdh* to *PRL6* (expressed comparable in control and cko) in old qPCR and half-life measurement experiments (line 725). So, we renewed the results of qPCR in Fig. 7G and dismiss the data of half-life measurement (because of the poor repeatability) (lines 1446-1450, picture A).

Symbol	log2FC	padj
Rpl6	0.0086091	0.957874
Gapdh	0.8859393	3.72E-09

Because the results of new sequencing data were unexpected and more aspects of defects were found, we are very sorry about the substantial change of the article. We tried our best to revise the manuscript and hoped to improve it and meet your requirements. We apologize so much if we cause you lots of trouble. Please don't hesitate to let me know if any actions were needed.

Once again, on behalf of my co-authors, thank you very much for your constructive comments and suggestions and kind help for improving our manuscript. We really appreciate your efforts and apologize to take up too much of your time. Looking forward to hearing from you.

Thank you and best regards!

The point-by-point response was listed below.

Reviewer #1 (Remarks to the Author):

In this manuscript by Tan et al., titled “The m6A reader PRRC2A is essential for successful completion of meiosis I in spermatogenesis by promoting mRNA stability”, the authors describe an essential role for PRRC2A during spermatogenesis. They demonstrate that PRRC2A is critical for spindle organization and chromosome segregation during metaphase I of meiosis. They further show that PRRC2A binds centrosome-encoding transcripts among others, and directly/indirectly affects their steady-state levels. The findings reported here are novel and experiments conducted are extensive. Few comments are listed below.

Authors' response: We are grateful for these very positive comments!

However, a substantial revision was made in the manuscripts with a huge change of conclusion. According to the suggestions of the editor and other reviewers, we performed RNA-seq and Ribo-seq using sorted spermatocytes and dismiss the RNA-seq and polysome profile using P20 testes. Based on the new sequencing data, our statement on the PRRC2A function was totally changed. We thought PRRC2A is not an mRNA stabilizer but potentially promotes the decay or translation of its mRNA targets, statements described detailedly below in major comment #2. We are very sorry about the substantial change to the article and we apologize so much that we cause you lots of trouble and take up too much of your time. Hope this revised manuscript can meet your requirements and look forward to your further suggestions.

Major comments:

1) The authors report that there is an MSCI defect in *Prrc2a* cKO based on increased signal intensity of Pol II at/near XY. While this is consistent with the upregulation of XY genes in whole testis RNA seq at P20 in *Prrc2a* cKO, other data do not fully support that the upregulation stems from an MSCI defect. MSCI defects lead to a mid-pachytene arrest (see Royo et al. 2010 *Curr Biol* or Turner 2015 *Annu Rev Genet*). The authors observe few TUNEL-positive pachytene cells consistent with this (Fig 1G), but this should be quantified and connected to the small (although not significant) increase in TUNEL staining seen at ~stages IV-VII (Fig 1H).

Authors' response: Thank you for these great suggestions! We have quantified the apoptosis of spermatocytes at different stages (Fig. S4B, lines 202-203), as you suggested and observed increased apoptosis in *Prrc2a*-cko mid-pachytene cells compared with controls. This might correspond to the slight increase in TUNEL signaling at stages IV-VII. Additionally, we found the number of apoptosis was increased starting from early pachytene (Fig. S4B, lines 212-214), this reflected other developmental defects during late prophase and was corresponding to the delayed progression from zygonema to pachynema (Fig. S4C, lines 209-214).

And MSCI defects lead to loss of gamma-H2AX accumulation on XY. This does not

appear to be the case in Fig 2A, or is this observed at a low frequency?

Authors' response: We checked our immunostaining images again and did not find any defect in gamma-H2AX signaling. We thought that the impaired MSCI in PRRC2A-null spermatocytes might be relatively mild or be a type of non-classical defect.

To test whether there is an MSCI defect, the authors would need to examine localization of silencing/MSCI factors on XY (e.g., gamma-H2AX; there are several others that could be tested although this is not necessary: HORMAD2, BRCA1, ATR, TOPBP1, MDC1).

Authors' response: Thank you for providing these important points! As you suggested, we detected ATR and MDC1 in spermatocyte spreads and observed normal expression patterns on XY chromosomes in PRRC2A-null cells (Fig. 2J, lines 203-206). For HORMAD2, TOPBP1, and MDC1, we tested several types of commercial antibodies but found they did not work, so we did not detect them. Considering no finding of abnormal expression for MSCI factors, we believed that the impaired MSCI may be caused by the downstream epigenetic abnormalities in the XY region, such as H3K9me3 (Hirota, T. et al. Developmental cell, 2018).

And/or the authors could perform RNA FISH of an XY gene to show that inappropriate gene XY expression is occurring specifically in pachytene cells.

Authors' response: We were very sorry that we did not have enough time to perform this experiment because of the Covid-19 pandemic. But the RNA-seq and qPCR results in Fig S6A&B may partly indicate impaired MSCI in PRRC2A-null pachytene spermatocytes.

2) The authors state that PRRC2A binds and stabilizes centrosome-encoding transcripts, and show a stabilization function for select targets by performing half-life measurements in cultured meiotic cells. For a more genome-wide examination, they examine the overlap between down regulated transcripts in Prrc2a cKO and PRRC2A RIP targets plus m6A-modified transcripts and detect ~380 transcripts. Do the upregulated transcripts show a smaller overlap, consistent with the principal function being to promote stability?

Authors' response: Thank you for providing these important points.

As you suggested, we tried to overlap the down-regulated and up-regulated DEGs with PRRC2A-m6A in our new sequencing data of sorted spermatocytes. We obtained 336 out of 1331 (25.24%) up-regulated DEGs and 75 out of 1389 (5.40%) down-regulated DEGs with PRRC2A-binding m6A-modified site (figure shown below, left panel). So, a greater overlap was observed between up-regulated DEGs and PRRC2A-m6A genes. However, the GO analysis of these overlapped genes cannot obtain the terms related to spermatocytes or cell division (figure shown below, right panels), so we thought this may not be helpful to the logic of the article, and we did not show these data in the revised manuscript.

Based on the sequencing data of sorted spermatocytes, we thought PRRC2A is not an mRNA stabilizer but potentially promotes the decay or translation of its mRNA targets.

Detailly, we found that PRRC2A downregulates mRNA abundance or upregulates translation efficiency of its targets at the transcriptome level (Fig. 5H, 362-372). Specifically, PRRC2A recognized spermatogonia-specific transcripts and downregulates their expression (Fig. 6A, 6G, S6E, lines 412-421). For genes involved in meiotic cell division, PRRC2A improves their translation efficiency (Fig. 7, lines 481-501). Considering that PRRC2A interacts with multiple proteins regulating mRNA metabolism and translation (YBX1, YBX2, PABPC1, FXR1, and EIF4G3) (Fig. 7H, S7B, lines 510-541), we believed that PRRC2A promotes mRNA degradation or enhance the translation. We were very sorry for the huge change in the statement of the PRRC2A function compared with the previous version of manuscript.

Similarly, the comparison made with centrosome-encoding transcripts excludes upregulated transcripts. How many centrosome-encoding transcripts overlap with the PRRC2A RIP plus m6A-modified plus upregulated transcripts? These comparisons should be explained in the text and contrasted with those already mentioned.

Authors' response: Thank you for these great suggestions. We tried to overlap the differentially expressed centrosome-encoding transcripts with PRRC2A RIP plus m6A-modified genes, but only few genes (combined up-regulated and down-regulated genes) were obtained in the new sequencing data for sorted spermatocytes (data not shown). So, we selected two representative genes (*Cep192*, and *Wnk1*) reported to regulate spindle assembly and chromosome alignment. We found that their protein abundance was decreased in PRRC2A-null spermatocytes (Fig. 7E, F), but the RNA abundance was not affected dramatically (Fig. 7G, lines 433-453). So, we speculate that PRRC2A may promote the translation of genes involved in meiotic cell division (lines 426-501).

Minor comments:

1) In lines 174-175, the authors state that XY asynapsis triggers impaired MSCI (and they cite a Mettl3 study). I am not sure what the authors mean by this, as the asynapsis of X and Y is connected to normal MSCI. Please could the authors check/clarify this statement and cite appropriate refs.

Authors' response: Thank you for bringing this up. We apologize for our rigorous and accurate expression and incorrect reference. According to previous studies, we replaced

the inappropriate statement by “Note that XY asynapsis and impaired meiotic sex chromosome inactivation (MSCI) frequently co-occur in defective spermatocytes, such as those from *Brdt*^{-/-} (Manterola, Marcia, et al. PLoS genetic, 2018) and *Raptor*^{-/-} (Xiong, Mengneng, et al. The FASEB Journal, 2017) mice” and citations were added correspondingly (lines 196-197).

2) Can the authors be certain that the interactions observed between PRCC2A and YBX1/YBX2 are not indirect through RNA, since interactions tested via IP are lost after RNase treatment (Fig 6C).

Authors' response: Thank you for these great suggestions. We repeated the IP experiment several times and indeed get the same results as shown in the manuscript. So, we are certain that PRRC2A interacts with YBX1/YBX2 indirectly and dependently on the existence of RNA.

This should be addressed in the text and the PRRC2A interaction statements should be modified accordingly.

Authors' response: We are very sorry for the unclear statement we made here. We have modified relevant descriptions in the manuscript as you suggested (lines 525-528).

3) Wojtas et al. Mol Cell 2017 reference should be included when testis m6A studies are described (line 70).

Authors' response: We are very sorry for our negligence. We have added the corresponding citation (line 77).

4) Figure S1B is hard to see. I suggest providing magnified insets.

Authors' response: Thank you for bringing this up. We have rearranged the pictures and provided magnified insets and hoped that it could make the image easier to watch (Fig. S1B).

5) In Fig 1H the annotation is probably meant to be $p=0.054$ and gamma-H2AX is incorrectly written throughout.

Authors' response: We are very sorry for our incorrect writing. We have corrected these in the revised manuscript (Fig. 1H) (“rH2AX” to “ γ H2AX”).

We apologized so much for our poor language and editing of our manuscript. We worked on the manuscript for a long time and the repeated addition and removal of sentences and sections obviously led to poor readability. We have now tried our best to work on both language and readability and hope that the flow and language level have been improved.

Reviewer #2 (Remarks to the Author):

The manuscript “The m6A reader PRRC2A is essential for successful completion of meiosis I in spermatogenesis by promoting mRNA stability” by Tan et al show that the m6A reader PRRC2A, using a male germ cell-specific *Prcc2a* knockout mouse model,

is necessary for meiosis I progression and propose that PRRC2A promotes the stability of transcripts of centrosome-contained proteins through binding them. This work aims to enhance our understanding of an important regulatory process (posttranscriptional regulation during meiosis I).

Authors' response: We are grateful for these positive comments!

However, a substantial revision was made in the manuscripts with a huge change of conclusion. According to your suggestions, we performed RNA-seq and Ribo-seq using sorted spermatocytes and dismiss the RNA-seq and polysome profile using P20 testes. Based on the new sequencing data, our statement on the PRRC2A function was totally changed. We thought PRRC2A is not an mRNA stabilizer but potentially promotes the decay or translation of its mRNA targets, statements described detailly below. We are very sorry about the substantial change to the article and we apologize so much that we cause you lots of trouble and take up too much of your time. Hope this revised manuscript can meet your requirements and look forward to your further suggestions.

Major concerns:

A lynchpin argument for this report is that “Volcano plot of RNA-seq data shows the transcriptome difference in P20 control and *Prrc2a*-cko testes. p-value = 0.01 and 1.2-fold expression difference are used as the cutoff” (Lines 1120-23) and that “we performed polysome analysis and found no significant difference in overall translation status between P20 *Prrc2a*-cko and control testes.” (Lines 328-329)

1) The experiments for RNA-seq and polysome analysis in this manuscript used whole testes. Thus, the data presented are not solid. This reviewer would like to require the authors use purified spermatogenic cells for RNA-seq and subsequent polysome analysis.

Authors' response: Thank you for providing these important points and we agreed with these ideas. As you suggested, we performed RNA-seq and Ribo-seq using sorted spermatocytes (lines 303-304, Fig 5, 6, 7A) for the revised manuscript. But because polysomes in purified male germ cells collapsed, we did not perform polysome profiling for sorted spermatocytes (Kang, Jun-Yan, et al. Science, 2022).

For the sequencing of spermatocytes, we initially planned to purify spermatocytes from P20 control and *Prrc2a*-cko testes using FACS, but we surprisedly found that late prophase spermatocytes were dramatically less in *Prrc2a*-cko P20 testes than in control P20 testes (Fig. S4C, lines 209-214), phenotypes we did not notice before. Moreover, the sequencing data revealed a quite different pattern of the transcriptome between using P20 testes and sorted spermatocytes (Fig. 5-7). Thus, we decided to dismiss the RNA-seq (lines 1413-1420, lines 1422-1444 (picture D-F), and lines 1536-1546 (picture A, B, E, F)) and polysome analysis (lines 1548-1558) data for P20 testes, and study functions of PRRC2A only based on the sequencing data for sorted spermatocytes.

2) The cutoff value for the expression difference is too low. This reviewer would like to require the authors to provide corresponding results in Figure 4 with a relative higher

cutoff value (at least, for example 1.5).

Authors' response: Thank you for bringing this up. According to your suggestion, we set more stringent cutoffs for significant difference which are fold change > 1.5 & p-adjusted < 0.05 for RNA-seq data, and fold change > 1.5 & $p < 0.05$ for Ribo-seq data (Fig. 5A-C), and the obtained genes were used for the subsequent analysis.

Minor issues:

1. The manuscript needs careful editing to improve the use of English grammar.

Authors' response: Thank you for bringing this up. We apologized so much for our poor language and editing of our manuscript. We worked on the manuscript for a long time and the repeated addition and removal of sentences and sections obviously led to poor readability. We have now tried our best to work on both language and readability and hope that the flow and language level have been improved.

2. The author indicated in Fig 2b that *Prrc2a*-cKO spermatocytes showed XY asynapsis in late-pachytene, but the MLH1 foci number in Fig S4F has no significant change. Could the authors present a probable explanation for these data?

Authors' response: Thank you for providing this important question. To answer this, we tried to count the MLH1 foci number in more pachytene spermatocytes and classify *PRRC2A*-null spermatocytes into two groups based on whether XY synapses or not (Fig. 2G, lines 193-195). We found that the number of MLH1 foci in *PRRC2A*-null spermatocytes with synapsed XY was normal but was decreased in spermatocytes with asynapsed XY (Fig. 2F, G). We thought that the reduced number could correspond to the XY asynapsis which may make the logic more reasonable and hoped to answer your concern.

Reviewer #3 (Remarks to the Author):

In this manuscript, Tan and colleagues explore the role of the m6A reader *PRRC2* in spermatogenesis. Using a powerful in vivo cKO mice model, authors demonstrate the role of this protein in the regulation of the correct chromosome alignment during meiosis. Using a combination of RNA sequencing and RIP-seq, MeRIP-seq authors propose a model in which the absence of *PRRC2* destabilize some of the centrosome-related mRNAs leading to chromosomal instability.

This manuscript shed light on *PRRC2* functions as an m6A reader in the spermatogenesis complex, linking the strong phenotype observed in these mice with the function of one m6A reader. The manuscript presents a strong dataset and provide and significant advance in the correct comprehension of the role of this protein in mRNA regulation. However, some of the affirmations need further analysis and or/controls to support the molecular mechanism they describe, and the link between m6A methylation and the phenotype observed should be demonstrated to a greater extent.

Authors' response: We are grateful for these very positive comments and good suggestions! However, a substantial revision was made in the manuscripts with a huge

change of conclusion. According to the suggestions of the editor and other reviewers, we performed RNA-seq and Ribo-seq using sorted spermatocytes and dismiss the RNA-seq and polysome profile using P20 testes. Based on the new sequencing data, our statement on the PRRC2A function was totally changed. We thought PRRC2A is not an mRNA stabilizer but potentially promotes the decay or translation of its mRNA targets, statements described detailly below. We are very sorry about the substantial change to the article and we apologize so much that we cause you lots of trouble and take up too much of your time. Hope this revised manuscript can meet your requirements and look forward to your further suggestions.

Major concerns

1. Deletion of PRRC2 seems to have a role in the stabilization of the transcripts analysis of the transcriptome reveals an important general transcript downregulation (Fig4). However differential expression analysis is not restrictive enough; a 1.2 fold change (\log_2 fold 0,26) might only reflect stochastic differences among the different replicates with no biological consequences. I understand the need of lowering the value of fold to detect small changes, otherwise screened because the analysis is made in complex tissues. But even in this case, variation among the replicates must be assessed and fold change adjusted accordingly to it in the differential expression analysis.

Authors' response: Thank you for providing these important points and we really appreciate your understanding. Considering the complexity of the whole testis tissue, we performed RNA-seq and Ribo-seq using sorted spermatocytes (lines 302-303) (also suggested by editor and other reviewer). And according to your suggestion, we set more stringent cutoffs for the significant difference which are fold change > 1.5 & p-adjusted < 0.05 for RNA-seq data, and fold change > 1.5 & $p < 0.05$ for Ribo-seq data (Fig. 5A-C), and the obtained genes were used for the subsequent analysis (shown in Fig. 5-7).

Does the protein level of the analyzed mRNAs change in p20 testes in cKO mice?

Authors' response: Because the cellular composition of P20 *Prrc2a*-cko testes was quite different from controls (dramatical less late prophase spermatocytes) (Fig. S4C, lines 209-214), we thought examining the protein levels in P20 testes may not be appropriate to explain the function of PRRC2A and we did not perform this detection. We are very sorry we did not answer your question directly.

2. Heatmaps showing X- and Y- linked genes seem to show a high variance in the heatmap with big differences among the replicates (Fig S6A), and in fact, some of the representative genes picked to confirm show no statistically significant increase or no biological significant increase. A wider panel of genes must be analyzed to confirm the general de-repression of X- and Y-linked genes.

Authors' response: Thank you for providing these important points and we understand your concern. As you suggested, we have examined more X- and Y-linked genes (15 genes) by qPCR (Fig S6B). And considering the complexity of the whole testis, we detected the expression level in sorted spermatocytes rather than in P20 testes (Fig S6B).

We hope these results could confirm the general de-repression of X- and Y-linked genes and answer your concern.

3. In figure 6D, stricter differential expression analysis is needed together with statistical significance of the overlap. Also, overlapping controls with the upregulated genes will be useful to discern if this overlap is significant or not.

Authors' response: Thank you for providing these important points. As you suggested, we tried to overlap the down-regulated and up-regulated DEGs with PRRC2A-m6A in our new sequencing data of sorted spermatocytes. We obtained 336 out of 1331 (25.24%) up-regulated DEGs and 75 out of 1389 (5.40%) down-regulated DEGs with PRRC2A-binding m6A-modified site (figure shown below, left panel). So, a greater overlap was observed between up-regulated DEGs and PRRC2A-m6A genes.

However, the GO analysis of these overlapped genes cannot obtain the terms related to spermatocytes or cell division (figure shown below, right panels), so we thought this may not be helpful to the logic of the article, and we did not show these data in the revised manuscript.

That concern can also be applied to Sup. figure 6E.

Authors' response: We tried to overlap the differentially expressed centrosome-encoding transcripts with PRRC2A RIP plus m6A-modified genes, but only few genes (combined up-regulated and down-regulated genes) were obtained in the new sequencing data for sorted spermatocytes (data not shown). So, we selected two representative genes (*Cep192*, and *Wnk1*) reported to regulate spindle assembly and chromosome alignment. We found that their protein abundance was decreased in PRRC2A-null spermatocytes (Fig. 7E, F), but the RNA abundance was not affected dramatically (Fig. 7G, lines 433-453). So, we speculate that PRRC2A may promote the translation of genes involved in meiotic cell division (lines 426-501).

4. Authors suggest that PRRC2 promote the stability of mRNAs recognizing m6A but only one-third of PRRC2 RIP-seq peaks overlaps m6A peaks,

Authors' response: This is a really important question. We thought that PRRC2A-RIP

may obtain some unspecific transcripts due to the limitation of the experimental accuracy like your comment in major concern #6. So, the overlap of PRRC2A RIP-seq peaks with m6A peaks may be relatively low (35.6%) (Fig. S6E). But, the distribution pattern of PRRC2A targets is similar to the patterns of m6A (Fig. 5D-F). The sequencing data also showed that PRRC2A-binding methylated transcripts showed a greater degree of regulation than PRRC2A-binding targets without m6A modifications. We thought these results confirm that PRRC2A exert functions as an m6A reader (Fig. 5I, lines 372-375) and may answer your concern.

and only 20% of those overlapping peaks are downregulated (in a low restrictive analysis). Some of the genes they analyze do not get downregulated despite having been immunoprecipitated in m6A and PRRC2 RIP (Cep152 and Cep192 Figures 6F and 6K & L made in p20 testis). Authors should justify these low occurrence numbers in the discussion and other possible effects that might be affecting the regulation of these genes,

Authors' response: Thank you for providing these important points. We analyzed the function of PRRC2A using the sequencing data of sorted spermatocytes. We thought PRRC2A is not an mRNA stabilizer but potentially promotes the decay or translation of its mRNA targets. We were very sorry for the huge change in the statement of the PRRC2A function compared with the previous version of manuscript.

Detailedly, we found that PRRC2A downregulates mRNA abundance or upregulates translation efficiency of its targets at the transcriptome level (Fig. 5H, 362-372). Specifically, PRRC2A recognized spermatogonia-specific transcripts and downregulates their expression (Fig. 6A, 6G, S6E, lines 412-421). For genes involved in meiotic cell division, PRRC2A improves their translation efficiency (Fig. 7, lines 481-501). Considering that PRRC2A interacts with multiple proteins regulating mRNA metabolism and translation (YBX1, YBX2, PABPC1, FXR1, and EIF4G3) (Fig. 7H, S7B, lines 510-541), we believed that PRRC2A promotes mRNA degradation or enhance the translation.

and if possible, demonstrate the link between m6A methylation and the downregulation of these genes.

Authors' response: As described above, we found PRRC2A-binding methylated transcripts showed higher RNA abundance and lower TE than PRRC2A-binding targets without m6A modifications, this result confirmed that PRRC2A exert functions as an m6A reader (Fig. 5I, lines 372-375).

5. PRCC2A RIP and M6A meRIP data (Figure 5K and L must be represented as % of input with proper controls (Dazl, GAPDH or m6A positive controls) in the same scale. Referencing data to IgG is not valid as it is a random low number and any reference to them will produce random results.

Authors' response: Thank you for bringing this up. According to your suggestions, we presented the PRCC2A RIP and M6A meRIP data with the percentage of input with the control of GAPDH in the revised manuscript (Fig. 7C, D).

6. Moreover, authors should consider performing CLIP assays to distinguish real interactions occurring in the cell from those just produced by biochemical affinity or by secondary interactions.

Authors' response: Thank you for the great suggestion. But we are sorry that we fail to perform CLIP assays for PRRC2A. We attempted to perform the assay in P20 PRRC2A-Flag testes according to the reported method (Xu, Qiushi, et al. *JoVE* (2019) and Vourekas, Anastassios, and Zissimos Mourelatos. Humana Press, 2014.), but no PRRC2A protein was pulled down by protein G beads conjugated with mouse anti-Flag antibody.

7. PRRC2A binds YBX1 and YBX2 in an RNA dependent fashion, suggesting they are bound to the same RNAs but not form a complex, however, PRRC2A binds PABPC1 in an RNA independent manner indicating they form part of a complex. Is it the stability effect the authors observe depending on the methylation of the RNA or is it due to the alteration of the interaction of PABPC1 with the RNA which will produce destabilization of the mRNA?

Authors' response: Thank you for this interesting and important question. As we answered in major concern #4, we thought PRRC2A regulates its targets in an m6A-dependent manner. But we also speculated PRRC2A may exert functions with the help of other cofactors (such as PABPC1 to promote stability) rather than regulating targets directly. We thought these two descriptions are not incompatible.

Because mouse PRRC2A is a large protein with 2157 aa (isoform 2, NCBI ID: NP_001185973.1) or 2158 aa (isoform 1, NCBI ID: NP_064411.2), besides the GRE domain (domain enriched with glycine, arginine, and glutamic acid; less than 648 aa) used to bind m6A modification (Wu, Rong, et al. *Cell research*, 2019), the function of its BAT-N domain and other large regions is basically unknown. In this paper, we PRRC2A interacts with multiple cofactors which are reported to regulate mRNA metabolism and/or promote translation initiation, including YBX1, YBX2, FXR1, PABPC1, and EIF4G3 (Fig. 7H, S7B, lines 510-541). Additionally, PRRC2A was localized in RNA granules (chromatoid body) (Fig. 1C), this indicated the possible phase separation functions of PRRC2A. Altogether, we propose that PRRC2A potentially act as a scaffold to recruit different transcript targets and corresponding cofactors in different developmental stages and locations, so that performs diverse regulations to its targets (lines 585-599).

Minor concerns

1. Some of the figures do not follow the text order making it a bit difficult to follow the logic. There are also some figure citations wrong placed in the text such as “figure 3I” in line 298.

Authors' response: We apologize for being careless. We have checked and corrected all the figure citations.

2. Is endogenous PRRC2A-FLAG less expressed than the wt? It seems despite the FLAG being more powerful antibody images are much more exposed. Ratios between Input IP with FLAG and PRRC2A antibodies are quite different.

Authors' response: Thank you for bringing this up. We found the protein abundance of PRRC2A is similar in wild-type and PRRC2A-FLAG testes (Fig. S5C). We thought the difference observed in previous figures (line 1521, picture A) may be due to the poor quality and inappropriate exposure, so we replaced the old version picture with new pictures in Fig. S5C and hoped to answer your concern.

REVIEWER COMMENTS

Reviewer #1 (Remarks to the Author):

In this revised manuscript titled "The m6A reader PRRC2A is essential for successful completion of meiosis I in spermatogenesis", the authors describe the role of PRRC2A during meiosis. They present a thorough description of the Prrc2a mutant phenotype and demonstrate defects during meiotic prophase and metaphase I. They perform PRRC2A RIP-seq along with RNA-seq and Ribo-seq in purified spermatocytes from Prrc2a mutants to decipher direct RNA targets and functions of PRRC2A. They show that RNAs with PRRC2A binding peaks that also contain m6A peaks tend to be upregulated in mutants and are normally highly expressed in spermatogonia, suggesting that PRRC2A downregulates spermatogonia RNA in spermatocytes. They also identify genes important for meiotic division that have PRRC2A binding peaks and decreased RPF counts in Prrc2a-null spermatocytes, indicating that PRRC2A potentially promotes translation of RNAs important for meiotic cell division. Consistent with this, IP-MS shows that PRRC2A interacts with several factors important for RNA metabolism in the testis. My concerns from the previous round of review have been addressed and I only have a few comments which are listed below:

Authors need to explain the RNA-seq and Ribo-seq experiments better in the main body of the text when the experiments are first described. What age mice were examined? What spermatocyte stages do they represent in wild type and in mutant? Were the mice assessed at an age prior to cell death/meiotic arrest disrupting the testis cell composition in mutants? Ie, can the wild type be fairly compared to the mutant? Also, the main body should explicitly state that spermatocytes were isolated and briefly describe the isolation method as all these points greatly affect the results/interpretation of data.

Is the following results section title correct? "PRRC2A deficiency causes decreased abundance and enhanced translation of its target mRNAs", since abstract states PRRC2A decreases abundance and improves translation efficiency.

Reviewer #2 (Remarks to the Author):

The authors have taken my suggestion which clarify the data and the presentation. However, two major issues need to be solved:

1. The authors stated "PRRC2A deficiency cause decreased abundance and enhanced translation efficiency of its target mRNA" (lines 297-298). It is puzzling that this statement indicated opposite effects on the final protein level of PRRC2A targets. How to explain this? The authors should give clear evidence to prove it.
2. The authors stated "PRRC2A recognizes spermatogonia-specific transcripts and downregulates their expression during meiotic prophase to promote the transition from mitosis to meiosis" (line 419-421). If this is a case, why differentiated spermatogonia and their progeny (such as preleptotene, leptotene, zygotene, etc.) developed well in the Prrc2a cKO mice?

Additional comments:

- 1) The title reads awkward, and may not be accurate because PRRC2A-null male germ cells, at least partial, can develop into spermatids (line 143).
- 2) Fig.1F and 1G do NOT support the statement "specifically nothing that most of the apoptotic cells were metaphase spermatocytes or round spermatids. Further, we found that the number of apoptotic cells was greatly increased in stage I-III tubules (which contain newly produced round spermatids)." (lines 160-162). The authors should give clear evidence for defining epithelial stages and

cell types.

3) The conclusion "PRRC2A-null prophase spermatocytes develop normally through the leptotene, zygotene, pachytene, diplotene stages, and the proportion of each type of spermatocytes was similar to controls" "we found autosomes in PRRC2A-null spermatocytes synapsed at the pachytene stage and separated at the diplotene stage as controls" is not solid, and may be wrong. The development of chromosome axes and the synaptonemal complex are usually used to define the substages of meiotic prophase I. Thus, spermatocyte nuclear spreading with immunostaining for SYCP3, the axial/lateral element (AE/LE) protein, and SYCP1, the central element (CE) protein of synaptonemal complex should be performed.

4) I am very curious to know how the authors can define the epithelial stages on the DAPI/pH3/ γ H2AX stained sections (Fig.3A and 3D). The authors have to provide clear evidence because this is a fundamental technique for the statement "PRRC2A deficiency leads to delayed metaphase entry and metaphase arrest" (line 215).

5) In mice, the first wave of spermatogenesis is, somehow, different from adult spermatogenesis, in particular at this case (Fig.S4C). The author should clarify it.

6) The authors declared that "Interestingly, we noticed genes required for meiotic recombination (such as Spo11 and Hormad1) were not affected by PRRC2A deficiency (Fig. 6E) which explained why PRRC2A-null spermatocytes were able to progress through meiotic prophase" (lines 402-405). Ccdc36, a component of DSB forming machinery, which is essential for meiosis, is "downregulated in PRRC2A-null spermatocytes" (lines 397-400). How to explain this?

Reviewer #3 (Remarks to the Author):

The authors made an extensive revision and answered satisfactorily to most of the major concerns, improving notably the quality and conclusions of the manuscript. Hence I recommend this article for publication after the correction of two minor concerns;

-Despite relative RNA abundance and translation is dependent on PRRC2A binding and m6A methylation (figure 5H and 5I) overlap of differentially translated/transcribed genes and genes bound to PRRC2A-m6A is relatively low (overlaps of Fig5A and Fig5G, figures sent to the reviewer). Authors should point out that in the discussion as most of these differentially expressed genes seem to be downstream of those directly regulated by PRRC2A

-Some of the footnotes still reflect $\text{Log}_2\text{FC} > 1.5$ when they refer to $\text{FC} > 1.5$, p .ex line 1352

REVIEWER COMMENTS

Reviewer #1 (Remarks to the Author):

In this revised manuscript titled “The m6A reader PRRC2A is essential for successful completion of meiosis I in spermatogenesis”, the authors describe the role of PRRC2A during meiosis. They present a thorough description of the *Prrc2a* mutant phenotype and demonstrate defects during meiotic prophase and metaphase I. They perform PRRC2A RIP-seq along with RNA-seq and Ribo-seq in purified spermatocytes from *Prrc2a* mutants to decipher direct RNA targets and functions of PRRC2A. They show that RNAs with PRRC2A binding peaks that also contain m6A peaks tend to be upregulated in mutants and are normally highly expressed in spermatogonia, suggesting that PRRC2A downregulates spermatogonia RNA in spermatocytes. They also identify genes important for meiotic division that have PRRC2A binding peaks and decreased RPF counts in *Prrc2a*-null spermatocytes, indicating that PRRC2A potentially promotes translation of RNAs important for meiotic cell division. Consistent with this, IP-MS shows that PRRC2A interacts with several factors important for RNA metabolism in the testis. My concerns from the previous round of review have been addressed and I only have a few comments which are listed below:

Authors' response: Thank you for your very positive comments! We are very honored to be recognized and affirmed by you. Your valuable suggestions really promote the quality of our research. We appreciate your careful review very much!

Authors need to explain the RNA-seq and Ribo-seq experiments better in the main body of the text when the experiments are first described. What age mice were examined? What spermatocyte stages do they represent in wild type and in mutant? Were the mice assessed at an age prior to cell death/meiotic arrest disrupting the testis cell composition in mutants? I.e., can the wild type be fairly compared to the mutant? Also, the main body should explicitly state that spermatocytes were isolated and briefly describe the isolation method as all these points greatly affect the results/interpretation of data.

Authors' response: Thank you for these important suggestions and we are very sorry for the unclear statement we made here. As you suggested, we have described some details of RNA-seq, Ribo-seq, and the isolation method of spermatocytes in the manuscript (lines 289-291 and lines 672-691). We performed RNA-seq and Ribo-seq using prophase spermatocytes (during leptotene, zygotene, pachytene, and diplotene stages which are before occurring of massive cell death and meiotic arrest during metaphase) from adult (P60) testes. Germ cells from adult *Prrc2a*-cko and control testes were stained with Hoechst 33342 and prophase spermatocytes were purified using FACS according to the previously reported method (doi:10.1002/cyto.a.22463 (2014)).

In adult *Prrc2a*-cko testes, the proportion of each type of prophase spermatocytes is similar to controls (Supplemental Fig. S4B, S5C). Therefore, we thought that the wild-type group could be fairly compared to the mutant group.

Is the following results section title correct? “PRRC2A deficiency causes decreased

abundance and enhanced translation of its target mRNAs”, since abstract states PRRC2A decreases abundance and improves translation efficiency.

Authors' response: We apologize so much for being careless. We have corrected this to “PRRC2A deficiency cause increased abundance and reduced translation efficiency of its target mRNA” in the manuscript (line 284). We are very sorry for making you confusing.

Reviewer #2 (Remarks to the Author):

The authors have taken my suggestion which clarify the data and the presentation. However, two major issues need to be solved:

Authors' response: We are very honored to be recognized and affirmed by you. Your valuable suggestions really promote the quality of our research. We appreciate your careful review very much!

1. The authors stated “PRRC2A deficiency cause decreased abundance and enhanced translation efficiency of its target mRNA” (lines 297-298). It is puzzling that this statement indicated opposite effects on the final protein level of PRRC2A targets. How to explain this? The authors should give clear evidence to prove it.

Authors' response: We apologize so much for being careless, this is incorrect writing. We mean “PRRC2A deficiency cause increased abundance and reduced translation efficiency of its target mRNA” and we have corrected this in the manuscript (line 284). We are very sorry for making you confusing.

2. The authors stated “PRRC2A recognizes spermatogonia-specific transcripts and downregulates their expression during meiotic prophase to promote the transition from mitosis to meiosis” (line 419-421). If this is a case, why differentiated spermatogonia and their progeny (such as preleptotene, leptotene, zygotene, etc.) developed well in the *Prrc2a* cKO mice?

Authors' response: Thank you for providing this important question! We have discussed this point in the discussion section (lines 488-499). We speculated that PRRC2A may not play a central role in differentiated spermatogonia and early-prophase spermatocytes. The expression of PRRC2A is low during these stages (Fig. 1B, C, S1) and some other proteins could also promote the transition from mitosis to meiosis, such as YTHDC2. Further, *Ccna2*, the typical gene whose overexpression results in impaired meiosis initiation, is expressed normally in PRRC2A-null spermatocytes (Fig. 4C, D). So, the development of differentiated spermatogonia and early-prophase spermatocytes may be less affected and could still develop well, although PRRC2A deficiency causes the abnormal upregulation of spermatogonia-specific genes. In contrary, during the late prophase, PRRC2A is highly expressed (Fig. 1B, C, S1) and exerts critical function, so multi-aspect defects occur in PRRC2A-null spermatocytes.

Additional comments:

1) The title reads awkward, and may not be accurate because PRRC2A-null male germ cells, at least partial, can develop into spermatids (line 143).

Authors' response: Thank you for bringing this up. Because of massive apoptosis and multi-aspect phenotypes of spermatids (Fig. 1F, Supplementary Figure S3A-C, Supplementary Figure S4A), we speculated spermatids may be produced with defects and their defects were the consequence of the impaired development of spermatocytes. So, only meiotic defects were emphasized in the title. But we also have changed the title to “The m6A reader PRRC2A is essential for meiosis I completion during spermatogenesis” to make the statement more fluent. Hope this could answer your concern.

2) Fig.1F and 1G do NOT support the statement “specifically nothing that most of the apoptotic cells were metaphase spermatocytes or round spermatids.

Authors' response: Thank you for providing this important point. We have quantified the apoptosis of metaphase spermatocytes and round spermatids and found significant increases in *Prrc2a*-cko testes (Supplementary Figure S4A, line 148).

Further, we found that the number of apoptotic cells was greatly increased in stage I-III tubules (which contain newly produced round spermatids).”(lines 160-162). The authors should give clear evidence for defining epithelial stages and cell types.

Authors' response: Thank you for your important suggestion and we understand your concern. We have described the detailed stage determination method for seminiferous tubules (lines 148-150 and lines 864-882) and showed the representative images (Supplementary Fig. S3C). In adult *Prrc2a*-cko testes, tubules with early-pachytene spermatocytes (cells close to the base of the lumen and with nuclear dense punctate γ H2AX signal) and round spermatids were determined as stage I-III; tubules with mid-pachytene spermatocytes (cells relatively far from the base of the lumen and with nuclear dense punctate γ H2AX signal) and round spermatids (little or no) were determined as stage IV-VII; tubules with leptotene spermatocytes (cells located at the base of the lumen and with nuclear γ H2AX signal), late-pachytene spermatocytes (cells far from the base of the lumen and had large nuclei with dense punctate γ H2AX signal) and no spermatid were determined as stage VIII-X; tubules with zygotene spermatocytes (cells located at the base of the lumen and with nuclear dense but not completely concentrated γ H2AX signal) and diplotene spermatocytes (cells far from the base of the lumen and had large nuclei with dense punctate γ H2AX signal) were determined as stage XI-XII. Only round and characteristic tubules were considered for followed analysis. Germ cells far from the base of the lumen and with condensed chromosomes aligning or aligned on the equatorial plate were determined as metaphase spermatocytes. Hope this could answer your concern.

3) The conclusion “PRRC2A-null prophase spermatocytes develop normally through the leptotene, zygotene, pachytene, diplotene stages, and the proportion of each type of spermatocytes was similar to controls” “we found autosomes in PRRC2A-null spermatocytes synapsed at the pachytene stage and separated at the diplotene stage as

controls” is not solid, and may be wrong. The development of chromosome axes and the synaptonemal complex are usually used to define the substages of meiotic prophase I. Thus, spermatocyte nuclear spreading with immunostaining for SYCP3, the axial/lateral element (AE/LE) protein, and SYCP1, the central element (CE) protein of synaptonemal complex should be performed.

Authors' response: Thank you for your great suggestion! As you suggested, we have also examined the expression of SYCP3 and SYCP1 on chromosome spreads (Supplementary Figure S4C, lines 168-172) and found the synapsis and separation of chromosomes were not affected in PRRC2A-null prophase spermatocytes. This result also supports our definition of the substages of meiotic prophase I using SYCP3 and γ H2AX.

4) I am very curious to know how the authors can define the epithelial stages on the DAPI/pH3/ γ H2AX stained sections (Fig.3A and 3D). The authors have to provide clear evidence because this is a fundamental technique for the statement “PRRC2A deficiency leads to delayed metaphase entry and metaphase arrest” (line 215).

Authors' response: Thank you for your question, as described in minor concern 2), we determined the stage of tubules based on the expression pattern of γ H2AX, the cell morphology, and the composition of germ cells. We have described the detailed stage determination method for seminiferous tubules (lines 148-150 and lines 864-882) and showed the representative images (Supplementary Fig. S3C).

5) In mice, the first wave of spermatogenesis is, somehow, different from adult spermatogenesis, in particular at this case (Fig.S4C). The author should clarify it.

Authors' response: Thank you for providing this important point. We have discussed this in the discussion section (lines 472-479). From phenotypes found in *Prrc2a*-cko testes, we thought the delayed entry and defective completion of metaphase could occur during both the first wave of spermatogenesis (Fig. 1I, 3B, 3C) and in adult spermatogenesis (Fig. 1H, 3D). However, the delayed progression from zygonema to pachynema only occur during the first wave of spermatogenesis (Supplementary Figure S4D), and the progression during meiotic prophase of adult spermatogenesis was normal in *Prrc2a*-cko testes (Fig. 2A, Supplementary Figure S4B).

6) The authors declared that “Interestingly, we noticed genes required for meiotic recombination (such as Spo11 and Hormad1) were not affected by PRRC2A deficiency (Fig. 6E) which explained why PRRC2A-null spermatocytes were able to progress through meiotic prophase” (lines 402-405). *Ccdc36*, a component of DSB forming machinery, which is essential for meiosis, is “downregulated in PRRC2A-null spermatocytes” (lines 397-400). How to explain this?

Authors' response: Thank you for the insightful question. We thought the extent of downregulation of *CCDC36* (\log_2 FC of RNA is -0.98, \log_2 FC of RPF is -0.81) may not be sufficient to cause defects. Or if the downregulation of *CCDC36* causes mild defects, some other proteins may also compensate for these consequences.

Reviewer #3 (Remarks to the Author):

The authors made an extensive revision and answered satisfactorily to most of the major concerns, improving notably the quality and conclusions of the manuscript. Hence I recommend this article for publication after the correction of two minor concerns;

Authors' response: Thanks for your very positive comments! We are very honored to be recognized and affirmed by you. Your valuable suggestions really promote the quality of our research. We appreciate your careful review very much!

-Despite relative RNA abundance and translation is dependent on PRRC2A binding and m6A methylation (figure 5H and 5I), overlap of differentially translated/transcribed genes and genes bound to PRRC2A-m6A is relatively low (overlaps of Fig5A and Fig5G, figures sent to the reviewer). Authors should point out that in the discussion as most of these differentially expressed genes seem to be downstream of those directly regulated by PRRC2A

Authors' response: Thank you for this important suggestion and we have discussed this point in the manuscript (lines 513-516). The sentences added were “Additionally, considering that many PRRC2A-binding transcripts do not contain m6A modification (2127 of 4081 PRRC2A-binding targets) (Fig. 5G), the large unknown regions of PRRC2A may also exert functions independent of m6A recognition to regulate mRNA targets directly.”

-Some of the footnotes still reflect $\text{Log}_2\text{FC} > 1.5$ when they refer to $\text{FC} > 1.5$, p .ex line 1352
Authors' response: We are very sorry for our incorrect writing. We have corrected this on lines 1249-1251.